# Lightweight Learner for Shared Knowledge Lifelong Learning

**Yunhao Ge**[1]                                                   *yunhaoge@usc.edu*
**Yuecheng Li**[1*]                                                *liyueche@usc.edu*
**Di Wu**[1*]                                                       *dwu92983@usc.edu*
**Ao Xu**[1*]                                                       *aoxu@usc.edu*
**Adam M. Jones**[2]                                               *adammj@usc.edu*
**Amanda Sofie Rios**[3]                                           *amanda.rios@intel.com*
**Iordanis Fostiropoulos**[1]                                      *fostirop@usc.edu*
**Shixian Wen**[4]                                                 *sx.wen@siat.ac.cn*
**Po-Hsuan Huang**[2]                                              *pohsuanh@usc.edu*
**Zachary William Murdock**[2]                                     *zmurdock@usc.edu*
**Gozde Sahin**[1]                                                 *gsahin@usc.edu*
**Shuo Ni**[1]                                                     *shuoni@usc.edu*
**Kiran Lekkala**[1]                                               *klekkala@usc.edu*
**Sumedh Anand Sontakke**[1]                                       *ssontakk@usc.edu*
**Laurent Itti**[1,2,5]                                            *itti@usc.edu*

[1] *Thomas Lord Department of Computer Science, University of Southern California*
[2] *Neuroscience Graduate Program, University of Southern California*
[3] *Intel Labs*
[4] *Shenzhen Institute of Advanced Technology, Chinese Academy of Sciences*
[5] *Dornsife Department of Psychology, University of Southern California*
[*] *Equal contribution as second author*

**Reviewed on OpenReview:** *https://openreview.net/forum?id=Jjl2c8kWUc*

## Abstract

In Lifelong Learning (LL), agents continually learn as they encounter new conditions and tasks. Most current LL is limited to a single agent that learns tasks sequentially. Dedicated LL machinery is then deployed to mitigate the forgetting of old tasks as new tasks are learned. This is inherently slow. We propose a new Shared Knowledge Lifelong Learning (SKILL) challenge, which deploys a decentralized population of LL agents that each sequentially learn different tasks, with all agents operating independently and in parallel. After learning their respective tasks, agents share and consolidate their knowledge over a decentralized communication network, so that, in the end, all agents can master all tasks. We present one solution to SKILL which uses Lightweight Lifelong Learning (LLL) agents, where the goal is to facilitate efficient sharing by minimizing the fraction of the agent that is specialized for any given task. Each LLL agent thus consists of a common task-agnostic immutable part, where most parameters are, and individual task-specific modules that contain fewer parameters but are adapted to each task. Agents share their task-specific modules, plus summary information ("task anchors") representing their tasks in the common task-agnostic latent space of all agents. Receiving agents register each received task-specific module using the corresponding anchor. Thus, every agent improves its ability to solve new tasks each time new task-specific modules and anchors are received. If all agents can communicate with all others, eventually all agents become identical and can solve all tasks. On a new, very challenging SKILL-102 dataset with 102 image classification tasks (5,033 classes in total, 2,041,225 training, 243,464 validation, and 243,464 test images), we achieve much higher (and SOTA) accuracy over 8 LL baselines, while also achieving near perfect parallelization. Code and data can be found at https://github.com/gyhandy/Shared-Knowledge-Lifelong-Learning

# 1 Introduction

Lifelong Learning (LL) is a relatively new area of machine learning (ML) research, in which agents continually learn as they encounter new tasks, acquiring novel task knowledge while avoiding forgetting of previous tasks (Parisi et al., 2019). This differs from standard train-then-deploy ML, which cannot incrementally learn without catastrophic interference across successive tasks (French, 1999).

Most current LL research assumes a single agent that sequentially learns from its own actions and surroundings, which, by design, is not parallelizable over time and/or physical locations. In the real world, tasks may happen in different places; for instance, we may need agents that can operate in deserts, forests, and snow, as well as recognize birds in the sky and fish in the deep ocean. The possibility of parallel task learning and sharing among multiple agents to speed up lifelong learning has traditionally been overlooked. To solve the above challenges, we propose a new Lifelong Learning challenge scenario, Shared Knowledge Lifelong Learning (SKILL): A population of originally identical LL agents is deployed to a number of distinct physical locations. Each agent learns a sequence of tasks in its location. Agents share knowledge over a decentralized network, so that, in the end, all agents can master all tasks. SKILL promises the following benefits: speedup of learning through parallelization; ability to simultaneously learn from distinct locations; resilience to failures as no central server is used; possible synergies among agents, whereby what is learned by one agent may facilitate future learning by other agents.

Application scenarios for SKILL include: 1) Users each take pictures of landmark places and buildings in their own city, then provide annotations for those. After learning and sharing, all users can identify all landmarks while traveling to any city. This could also apply to recognizing products in stores or markets in various countries, or foods at restaurants worldwide. Thus, even though each teacher only learns at one or a few locations (or tasks), eventually all users may be interested in knowledge from all locations, as it will be useful during travel. 2) Agents in remote outposts worldwide with limited connectivity are taught to recognize symptoms of new emerging diseases, then share their knowledge to allow any agent to quickly identify all diseases. 3) Explorers are dispatched to various remote locations and each learns about plant or animal species they encounter, then later shares with other agents who may encounter similar or different species. 4) Each time a criminal of some sorts is apprehended (e.g., shoplifter, insurgent, spy, robber, sex offender, etc), the local authorities take several hundred pictures to learn to identify that person. Then all local authorities share their knowledge so that any criminal can later be identified anywhere.

However, to solve SKILL, one must address the following challenges:

Chal-1 **Distributed, decentralized learning of multiple tasks.** A solution to SKILL should support a population of agents deployed over several physical locations and each learning one or more sequential tasks. For resilience reasons, the population should not rely on a single central server.

Chal-2 **Lifelong learning ability**: Each agent must be capable of lifelong learning, i.e., learning a sequence of tasks with minimal interference and no access to previous data as each new task is learned.

Chal-3 **Shareable knowledge representation:** The knowledge representation should easily be shared and understood among agents. Agents must be able to consolidate knowledge from other agents in a decentralized, distributed fashion.

Chal-4 **Speedup through parallelization:** Shared knowledge should be sufficiently compact, so that the benefits from using multiple parallel agents are not erased by communications costs. Adding more agents should result in greater speedup compared to a single agent. We measure speedup as the the ratio of time it takes for one agent to learn all tasks compared to $N$ agents (larger is better). As a goal for our work, we strive for a speedup of at least $0.5 \times N$ with $N$ agents, where perfect speedup would be $1.0 \times N$ if there was no parallelization and communications overhead.

Chal-5 **Ability to harness possible synergies among tasks:** When possible, learning some tasks may improve learning speed or performance at other, related tasks.

To address the SKILL challenge, we take inspiration from neuroscience. Many approaches to LL involve at least partially retraining the core network that performs tasks (feature extraction backbone plus classification head), as every new task is learned. But transmitting and then merging these networks across multiple agents would incur very high communications and computation costs. With the exception of perceptual learning, where human visual cortex may indeed be altered when learning specific visual discrimination tasks for days

or weeks (Goldstone, 1998; Dosher & Lu, 2017), there is little evidence that our entire visual cortex — from early stage filters in primary visual cortex to complex features in inferotemporal cortex — is significantly altered when we just learn, e.g., about a new class of flowers from a few exemplars. Instead, the perirhinal cortex (and more generally the medial temporal lobe) may be learning new representations for new objects by drawing upon and combining existing visual features and representations from visual cortex (Deshmukh et al., 2012). This may give rise to specialized "grandmother cells" (Bowers, 2017) (or Jennifer Aniston neurons; Quiroga et al. (2005); Quiroga (2017)) that can be trained on top of an otherwise rather immutable visual cortex backbone. While the grandmother cell hypothesis remains debated in neuroscience (vs. distributed representations; Valdez et al. (2015)), here, it motivates us to explore the viability of a new lightweight lifelong learning scheme, where the feature extraction backbone and the latent representation are fixed, and each new object class learned is represented by a single new neuron that draws from this representation.

From this inspiration, we propose a simple but effective solution to SKILL based on new lightweight lifelong learning (LLL) agents. Each LLL agent uses a common frozen backbone built-in at *initialization*, so that only the last layer *(head)* plus some small adjustments to the backbone *(beneficial biases)* are learned for each task. To eliminate the need for a task oracle, LLL agents also learn and share summary statistics about their training datasets, or share a few training images, to help other agents assign test samples to the correct head (task mapper). On a new, very challenging dataset with 102 image classification tasks (5,033 classes in total, 2,041,225 training, 243,464 validation, and 243,464 test images), we achieve higher accuracy compared to 8 LL baselines, and also near-perfect parallelization speedup.

Our main contributions are: (1) We formulate a new Lifelong learning challenge, Shared Knowledge Lifelong Learning (SKILL), which focuses on parallel (sped up) task learning and knowledge sharing among agents. We frame SKILL and contrast it with multi-task learning, sequential LL, and federated learning (Sec. 3). (2) A new LL benchmark dataset: SKILL-102, with 102 complex image classification tasks. To the best of our knowledge, it is the most challenging benchmark to evaluate LL and SKILL algorithms in the image classification domain, with the largest number of tasks, classes, and inter-task variance (Sec. 4). (3) A solution to the SKILL problem: Lightweight Lifelong Learning (LLL) for efficient knowledge sharing among agents, using a fixed shared core plus task-specific shareable modules. The need for a task oracle is eliminated by using a task mapper, which can automatically determine the task at inference time from just an input image (Sec. 5). (4) Our SKILL algorithm achieves SOTA performance on three main metrics: High LL task accuracy (less catastrophic forgetting), low shared (communication) resources, and high speedup ratio (Sec. 6). (5) The proposed Lightweight Lifelong Learner shows promising forward knowledge transfer, which reuses the accumulated knowledge for faster and more accurate learning of new tasks.

## 2 Related Works

### 2.1 Lifelong Learning

Lifelong Learning (LL) aims to develop AI systems that can continuously learn to address new tasks from new data, while preserving knowledge learned from previously learned tasks (Masana et al., 2022). It also refers to the ability to continually learn over time by accommodating new knowledge while retaining previously learned experiences (Parisi et al., 2019). LL is challenging because it is usually assumed that the training data from previous tasks is not available anymore while learning new tasks; hence one cannot just accumulate training data over time and then learn from all the data collected so far. Instead, new approaches have been proposed, which fall under three main branches (De Lange et al., 2021): *(1) Regularization methods* add an auxiliary loss term to the primary task objective to constrain weight updates, so as to minimally disturb previously learned knowledge while learning new tasks. The extra loss can be a penalty on the parameters (EWC (Kirkpatrick et al., 2017), MAS (Aljundi et al., 2018) and SI (Zenke et al., 2017)) or on the feature-space (FDR (Benjamin et al., 2018)), such as using Knowledge Distillation (LwF (Li & Hoiem, 2017), DMC (Zhang et al., 2020)). *(2) Parameter-Isolation methods* assign a fixed set of model parameters to a task and avoid over-writing them when new tasks are learned (SUPSUP (Wortsman et al., 2020)), PSP (Cheung et al., 2019) and BPN (Wen et al., 2021). *(3) Rehearsal methods* use a buffer containing sampled training data from previous tasks, as an auxiliary to a new task's training set. The buffer can be used either at the end of the task training (iCaRL, ER (Rebuffi et al., 2017b; Robins, 1995)) or during training (GSS, AGEM,

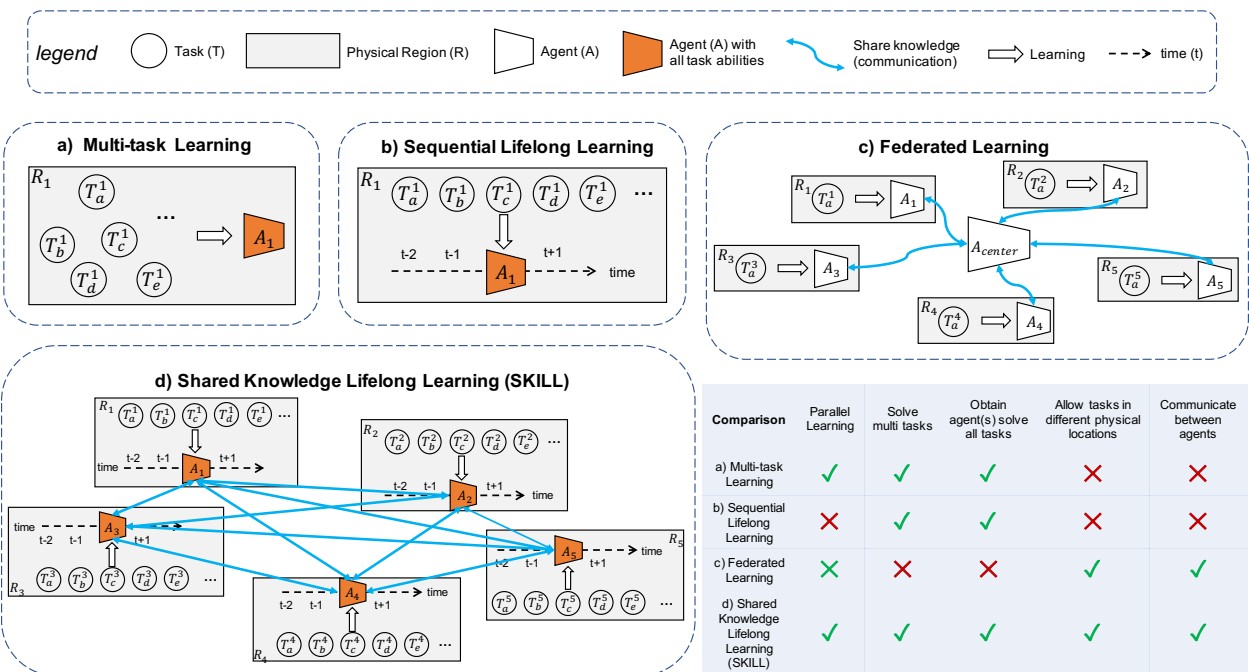

Figure 1: SKILL vs. related learning paradigms. a) Multi-task learning (Caruana, 1997): one agent learns all tasks at the same time in the same physical location. b) Sequential Lifelong Learning (S-LL) (Li & Hoiem, 2017): one agent learns all tasks sequentially in one location, deploying LL-specific machinery to avoid task interference. c) Federated learning (McMahan et al., 2017): multiple agents learn *the same task* in different physical locations, then sharing learned knowledge (parameters) with a center agent. d) Our SKILL: different S-LL agents in different physical regions each learn tasks, and learned knowledge is shared among all agents, such that finally all agents can solve all tasks. Bottom-right table: Strengths & weaknesses of each approach.

AGEM-R, GSS, DER, DERPP (Lopez-Paz & Ranzato, 2017; Chaudhry et al., 2018; Aljundi et al., 2019; Buzzega et al., 2020)). However, most traditional LL algorithms cannot satisfy the requirement of SKILL: parallel learning for speeding up, and sharing knowledge among agents.

## 2.2 Multi-task Learning

Multi-Task Learning (MTL) aims to leverage useful information contained in multiple related tasks to help improve the generalization performance of all the tasks (Zhang & Yang, 2021; Crawshaw, 2020; Ruder, 2017). The main difference between MTL and SKILL is that MTL assumes that all tasks are located in the same physical region, and that one can access the datasets of all tasks at the same time (Zhang & Yang, 2021). While MTL learns multiple tasks together, SKILL assumes that different knowledge sources are separated in different physical regions and different agents should learn them in parallel.

## 2.3 Federated Learning

Federated learning (FL) is a machine learning setting where many clients (e.g., mobile devices, networked computers, or even whole organizations) collaboratively train a model under the orchestration of a central server, while keeping the training data decentralized Kairouz et al. (2021); Li et al. (2020); Bonawitz et al. (2019). As shown in Fig. 1, compared with SKILL: (1) FL agents usually learn the *same task* from multiple partial datasets in different locations, relying on the central server to accumulate and consolidate the partial knowledge provided by each agent. In contrast, SKILL agents solve different tasks, and share knowledge over a decentralized network. (2) Each SKILL agent may learn multiple tasks in sequence, and hence must solve the LL problem of accumulating new knowledge while not forgetting old knowledge. Sequences of tasks

and the LL problem are usually not a primary focus of FL, with a few exceptions Yoon et al. (2021) which still do not directly apply to SKILL, as they focus on a single task for all agents and use a central server. Because federated learning relies on a central server, it is susceptible to complete failure if that server is destroyed; in contrast, in SKILL, as long as not all agents are destroyed, the surviving agents can still share and master some of the tasks.

### 2.4 Other methods that may help solve SKILL

One related direction is to share a compact representation dataset: Dataset distillation Wang et al. (2018) combines all training exemplars into a small number of super-exemplars which, when learned from using gradient descent, would generate the same gradients as the larger, original training set. However, the distillation cost is very high, which could not satisfy the $0.5N$ speedup requirement. Another related direction is to reuse shared parameters for different tasks: Adversarial reprogramming Elsayed et al. (2018) computes a single noise pattern for each task. This pattern is then added to inputs for a new task and fed through the original network. The original network processes the combined input + noise and generates an output, which is then remapped onto the desired output domain. However, the cost of the reprogramming training process is high, which could not satisfy the $0.5N$ speedup requirement. Another related direction is to use a generative adversarial network (GAN Goodfellow et al. (2020)) to learn the distribution of different datasets and generate for replay. Closed-loop GAN (CloGAN Rios & Itti (2018)) could be continuously trained with new data, while at the same time generating data from previously learned tasks for interleaved training. However, the GAN needs more parameters to transmit, and the high training time does not satisfy the $0.5N$ speedup requirement.

## 3 Shared knowledge in lifelong learning (SKILL)

The chief motivation for SKILL is to enable the next generation of highly-efficient, parallelizable, and resilient lifelong learning.

**Assumptions:** (1) A population of $N$ agents wants to learn a total of $T$ different tasks separated into $N$ physical regions. (2) Each agent $i$ asynchronously learns $1 \leq T_i \leq T$ tasks, in sequence, from the distinct inputs and operating conditions it encounters. As in standard LL, training data from previous tasks is not available anymore while learning the next task. (3) Each agent performs as a *"teacher"* for its $T_i$ tasks, by sharing what it has learned with the other $N-1$ agents; at the same time, each agent also performs as a *"student"* by receiving knowledge from the other $N-1$ agents. In the end, every agent has the knowledge to solve all $T$ tasks. Fig. 1 contrasts SKILL with other learning paradigms. Note how here we use "teacher" and "student" to distinguish the two roles that every agent will perform; this is different from and not to be confused with other uses of student/teacher terminology, for example in knowledge distillation. (4) There is a perfect task oracle at training time, i.e., each agent is told which tasks it should learn. (5) There is a clear separation between tasks, and between training and test phases.

**Evaluation metrics: (1) CPU/computation expenditure.** This metric is important to gauge the efficacy of an approach and its ability to scale up with more agents operating in parallel. Wall-clock time is the main metric of interest, so that speedup can be achieved through parallelism. Thus, if $N$ agents learn for 1 unit of time, wall-clock time would be 1, which is an $N$-fold speedup over a single sequential agent. In practice, speedup $< N$ is expected because of overhead for sharing, communications, and knowledge consolidation. Because wall clock time assumes a given CPU or GPU speed, we instead report the number of multiply-accumulate (MAC) operations. **(2) Network/communication expenditure.** Sharing knowledge over a network is costly and hence should be minimized. To relate communications to computation, and hence allow trade-offs, we assume a factor $\alpha = 1,000$ MACs / byte transmitted. It is a hyperparameter in our results that can be easily changed to adapt to different network types (e.g., wired vs. wireless). **(3) Performance:** After the population of $N$ agents has collectively learned all $T$ tasks, we report aggregated (averaged) performance over all $T$ tasks (correct classification rate over all tasks). Note how here we assume that there is *no task oracle at test time.* After training, agents should be able to handle any input from any task without being told which task that input corresponds to. SKILL does not assume a free task oracle because transmitting

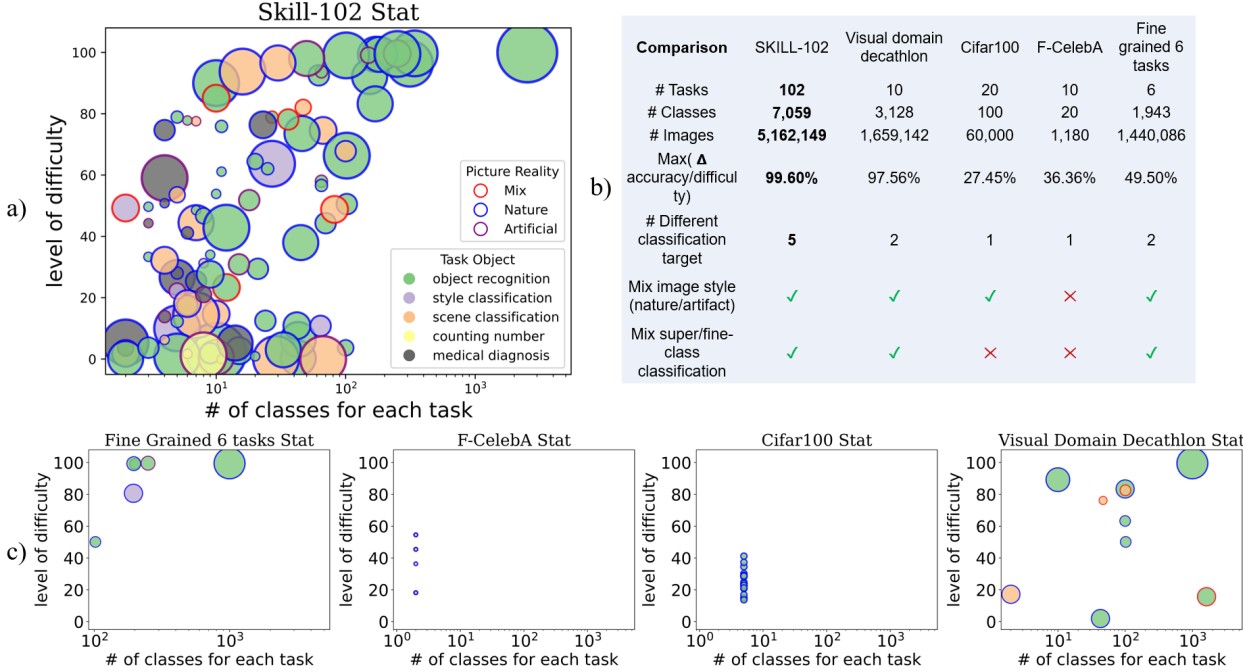

Figure 2: (a) SKILL-102 dataset visualization. Task difficulty (y-axis) was estimated as the error rate of a ResNet-18 trained from scratch on each task for a fixed number of epochs. Circle size reflects dataset size (number of images). (b) Comparison with other benchmark datasets including Visual Domain Decathlon (Rebuffi et al., 2017a), Cifar-100 (Krizhevsky et al., 2009), F-CelebA (Ke et al., 2020), Fine-grained 6 tasks (Russakovsky et al., 2014) (Wah et al., 2011), (Nilsback & Zisserman, 2008b), (Krause et al., 2013), (Saleh & Elgammal, 2015), (Eitz et al., 2012) c) Qualitative visualization of other datasets, using the same legend and format as in a).

training data across agents is potentially very expensive. Thus, agents must also share information that will allow receiving agents to know when a new test input relates to each received task.

**Open questions:** *What knowledge should be shared?* SKILL agents must share knowledge that is useful to other agents and avoid sharing local or specialized knowledge that may be misleading, in conflict with, or inappropriate to other agents. The shared knowledge may include model parameters, model structure, generalizations/specializations, input data, specific contextual information, etc. There are also size/memory/communication constraints for the shared knowledge. *When and how to share?* Different communication network topologies and sharing frequencies likely would lead to different results. Here, we will sidestep this problem and assume a fully connected communication network, and broadcast sharing from each agent to all others each time a new task has been learned.

## 4 SKILL-102 dataset

We use image classification as the basic task framework and propose a novel LL benchmark dataset: SKILL-102 (Fig. 2). SKILL-102 consists of 102 image classification datasets. Each one supports one complex classification task, and the corresponding dataset was obtained from previously published sources (e.g., task 1: classify flowers into 102 classes, such as lily, rose, petunia, etc using 8,185 train/val/test images (Nilsback & Zisserman, 2008a); task 2: classify 67 types of scenes, such as kitchen, bedroom, gas station, library, etc using 15,524 images (Quattoni & Torralba, 2009); full dataset sequence and details in Suppl. Fig. S5.

In total, SKILL-102 is a subset of all datasets/tasks and images in DCT, and comprises 102 tasks, 5,033 classes and 2,041,225 training images (Suppl. Sec. A and Suppl. Fig. S5). After training, the algorithm is presented 243,464 test images and decides, for each image, which of the 5,033 classes it belongs to (no task oracle). To the best of our knowledge, SKILL-102 is the most challenging completely real (not synthesized or

permuted) image classification benchmark for LL and SKILL algorithms, with the largest number of tasks, number of classes, and inter-task variance.

## 5 Lightweight Lifelong Learner for SKILL

To satisfy the requirements of SKILL (see Introduction), we design Lightweight Lifelong Learning (LLL) agents. The design motivation is as follows: We propose to decompose agents into a generic, pretrained, common representation backbone endowed into all agents at manufacturing time, and small task-specific decision modules. This enables distributed, decentralized learning as agents can learn their own tasks independently (Chal-1). It also enables lifelong learning (Chal-2) in each agent by creating a new task-specific module for each new task. Because the shared modules are all operating in the common representation of the backbone, this approach also satisfies (Chal-3). Using compact task-specific modules also aims to maximize speedup through parallelization (Chal-4). Finally, we show a few examples where knowledge from previously learned tasks may both accelerate the learning and improve the performance on new tasks (Chal-5).

Fig. 3 shows the overall pipeline and 4 roles for each agent. Agents use a common frozen backbone and only a compact task-dependent "head" module is trained per agent and task, and then shared among agents. This makes the cost of both training and sharing very low. Head modules simply consist of (1) a classification layer that operates on top of the frozen backbone, and (2) a set of beneficial biases that provide lightweight task-specific re-tuning of the backbone, to address potentially large domain gaps between the task-agnostic backbone and the data distribution of each new task. To eliminate the need for a task oracle, LLL agents also learn and share task anchors, in the form of summary statistics about their training datasets, or share a few training images, to help other agents assign test samples to the correct head at test time (task mapper). Two representations for task anchors, and the corresponding task mapping mechanisms, are explored: Gaussian Mixture Model Classifier (GMMC) and Mahalanobis distance classifier (MAHA). Receiving agents simply accumulate received heads and task anchors in banks, and the anchors for all tasks received so far by an agent are combined to form a task mapper within that agent. We currently assume a fully connected communication network among all agents, and every agent, after learning a new task, broadcasts its head and task anchor to all other agents. Hence, all agents become identical after all tasks have been learned and shared, and they all can master all tasks. At test time, using one of all identical agents, we first run input data through the task mapper to recover the task, and then invoke the corresponding head to obtain the final system output. The task mapper eliminates the need for a task oracle at test time. The combination of using a pre-trained backbone, task-specific head and BB, and task mapper enables lifelong learning in every agent with minimal forgetting as each agent learns a sequence of tasks (see results).

**Pretrained backbone:** We use the xception (Chollet, 2017) pretrained on ImageNet (Deng et al., 2009), as it provides a good balance between model complexity and expressivity of the embedding. The backbone is embedded in every agent at manufacturing time and is frozen. It processes $299 \times 299$ RGB input images, and outputs a 2048D feature vector. Any other backbone could be used, depending on available resources.

**Beneficial Biases:** To address potentially large domain shifts between ImageNet and future tasks (e.g., line-drawing datasets, medical imaging datasets, astronomy datasets), we designed beneficial biases (BB). Inspired by the Beneficial Perturbation Network (BPN) of Wen et al. (2021), BB provides a set of task-dependent, out-of-network bias units which are activated per task. These units take no input. Their constant outputs add to the biases of the neurons already present in the backbone network; thus, they provide one bias value per neuron in the core network. This is quite lightweight, as there are far fewer neurons than weights in the backbone (22.9M parameters but only 22k neurons in xception). Different from BPN, which works best in conjunction with an LL method like EWC (Kirkpatrick et al., 2017) or PSP (Cheung et al., 2019), and only works on fully-connected layers, BB does not require EWC or PSP, and can perform as an add-on module on both convolutional layers (Conv) and fully-connected layers (FC). Specifically, for each Conv layer, we have

$$y = Conv(x) + b + B \tag{1}$$

with input feature $x \in \mathbb{R}^{w*h*c}$, output feature $y \in \mathbb{R}^{w'*h'*c'}$. $b \in \mathbb{R}^{c'}$ is the original frozen bias of the backbone, and $B \in \mathbb{R}^{c'}$ is our learnable beneficial bias. The size of $B$ is equal to the number of kernels ($c'$) in this Conv layer. ($w, h, c$ and $w', h', c'$ denote the width, height and channels of the input and output feature

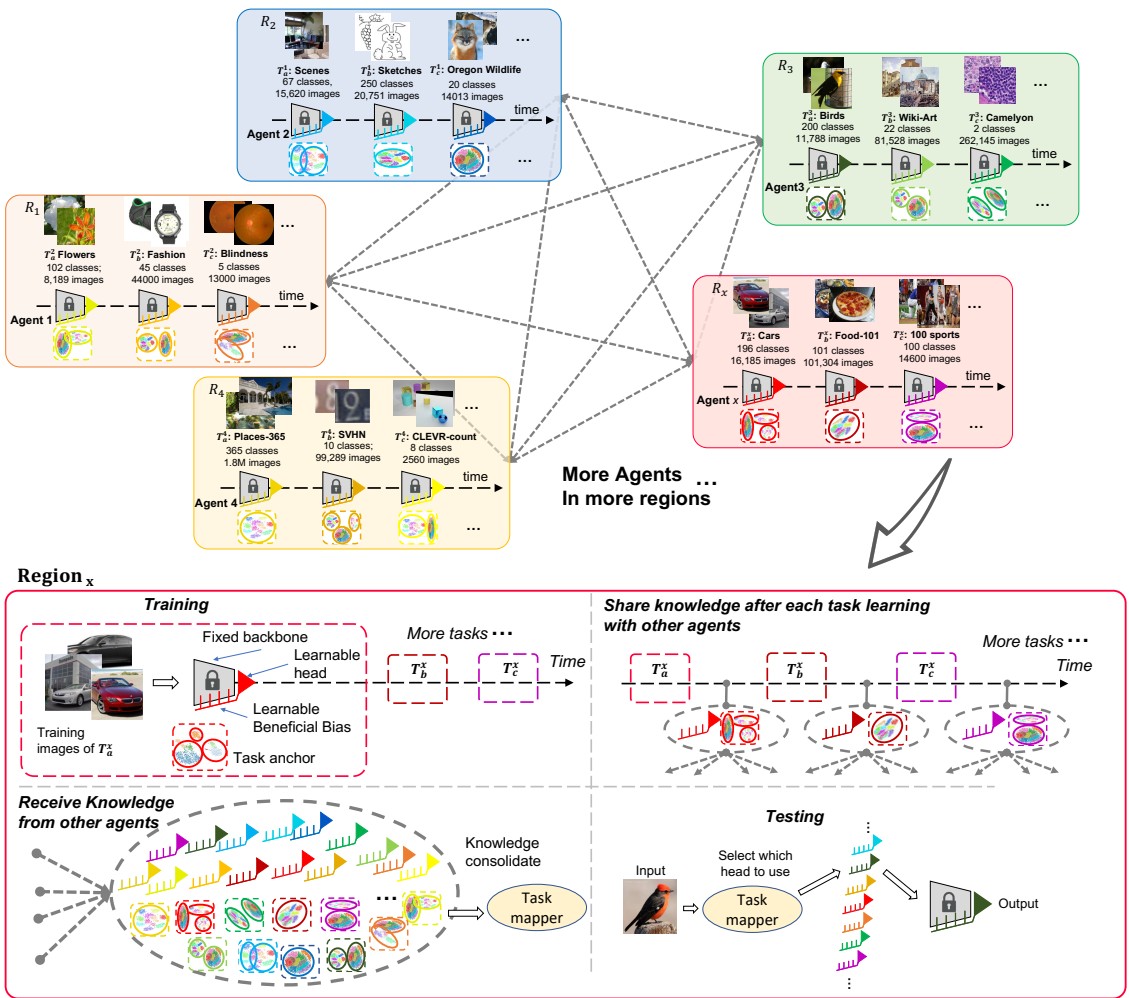

Figure 3: Algorithm design. Top: overall pipeline, where agents are deployed in different regions to learn their own tasks. Subsequently, learned knowledge is shared among all agents. Bottom: Zoom into the details of each agent, with 4 main roles: **1) Training:** agents use a common pre-trained and frozen backbone, stored in ROM memory at manufacturing time (gray trapezoid with lock symbol). The backbone allows the agent to extract compact representations from inputs (e.g., with an xception backbone, the representation is a latent vector of 2048 dimensions, and inputs are $299 \times 299$ RGB images). Each agent learns a task-specific head (red triangle) for each new task. A head consists of the last fully-connected layer of the network plus our proposed LL beneficial biasing units (BB) that provide task-dependent tuning biases to all neurons in the network (one float number per neuron). During training, each agent also learns a GMMC or Mahalanobis task anchor which will form a task mapper. **2) Share knowledge** with other agents: each agent shares the learned task-specific head, Beneficial Bias (BB), and GMMC module (or training images for Mahalanobis) with all other agents. **3) Receive knowledge** from other agents: each agent receives different heads and GMMC/Mahalanobis task mapper anchors from other agents. All heads are stored in a head bank and all task anchors are consolidated to form a task mapper. **4) Testing:** At test time, an input is first processed through the task mapper. This outputs a task ID, used to load up the corresponding head (last layer + beneficial biases) from the bank. The network is then equipped with the correct head and is run on the input to produce an output.

maps respectively.) For FC layers,

$$y = FC(x) + b + B \tag{2}$$

with $x \in \mathbb{R}^l$, $y \in \mathbb{R}^{l'}$, $b \in \mathbb{R}^{l'}$ and $B \in \mathbb{R}^{l'}$. The size of $B$ (beneficial bias) is equal to the number of hidden units ($l'$) in this FC layer.

**GMMC task mapper:** To recover task at test time, each agent also learns Gaussian Mixture clusters (GMMC) (Rios & Itti, 2020) that best encompass each of its tasks' data, and shares the cluster parameters (means + diagonal covariances). This is also very fast to learn and very compact to share. As shown in Fig. 3(bottom right), during training, each agent clusters its entire training set into $k$ Gaussian clusters:

$$f(x) = \sum_{i=1}^{k} \phi_i \mathcal{N}(x|\mu_i, \Sigma_i), \qquad \sum_{i=1}^{k} \phi_i = 1 \tag{3}$$

We use $k = 25$ clusters for every task (ablation studies in Appendix). In sharing knowledge, each agent performs a "teacher" role on its learned task and shares the mean and diagonal covariance of its clusters with all other agents (students). In receiving knowledge, each agent performs a "student" role and just aggregates all received clusters in a bank to form a task mapper with $kT$ clusters, keeping track of which task any given cluster comes from: $D_{map}() = \{(\mathcal{N}_1, \phi_1) : 1, ..., (\mathcal{N}_{kT}, \phi_{kT}) : T\}$. At test time, a image $x_i$ is evaluated against all clusters received so far, and the task associated with the cluster closest to the test image is chosen: $Task = D_{map}((\mathcal{N}_m, \phi_m))$, where $m = \arg\max_m(P(m, x_i))$. The probability of image $x_i$ belonging to the $m^{th}$ Gaussian cluster is given by:

$$P(m, x_i) = \frac{\phi_m \mathcal{N}(x|\mu_m, \Sigma_m)}{\sum_{n=1}^{kT} \phi_n \mathcal{N}(x|\mu_n, \Sigma_n)} \tag{4}$$

**Mahalanobis task mapper:** To perform as a task mapper, the Mahalanobis distance (MAHA) method (Lee et al., 2018) learns $C$ class-conditional Gaussian distributions $\mathcal{N}(x|\mu_c, \hat{\Sigma})$, $c = 1,2, ... C$, where $C$ is the total number of classes of all $T$ tasks and $\hat{\Sigma}$ is a tied covariance computed from samples from all classes. The class mean vectors and covariance matrix of MAHA are estimated as: $\mu_c = \frac{1}{N_c} \sum_{i:y_i=c} x_i$ ($N_c$: number of images in each class) and $\hat{\Sigma} = \frac{1}{N} \sum_{c=1}^{C} \sum_{i:y_i=c} (x_i - \mu_c)(x_i - \mu_c)^T$, ($N$: total number of images shared to the student agent). In training, each teacher agent computes the mean of each class within its task and randomly samples a variable number $m$ of images per class. In our experiments, we use $m = 5$ images/class for every task. During sharing knowledge, each agent shares the sample class means along with the saved images with all other agents. The shared images received by the student agents are used to compute the tied covariance. Similar to GMMC, the student agents also maintain a task mapper to keep track of which task any given class comes from. For a test image $x$, MAHA computes the Mahalanobis distance for all classes received so far and assigns the test image to the task associated with the smallest Mahalanobis distance, defined as:

$$\arg\min_c (x - \mu_c)^T \hat{\Sigma}^{-1} (x - \mu_c) \tag{5}$$

**System implementation details:** (1) Frozen xception backbone (Chollet, 2017), with 2048D latent representation. (2) Each agent learns one "head" per task, which consists of one fully-connected layer with 2048 inputs from the backbone and $c$ outputs for a classification task with $c$ classes (e.g., task 1 is to classify $c = 102$ types of flowers), and BB biases that allow us to fine-tune the backbone without changing its weights, to mitigate large domain shifts. (3) Each agent also fits $k = 25$ Gaussian clusters in the 2048D latent space to its training data. (4) At test time, a test image is presented and processed forward through the xception backbone. The GMMC classifier then determines the task from the nearest Gaussian cluster. The corresponding head is loaded and it produces the final classification result: which image class (among 5,033 total) the image belongs to. (5) The workflow is slightly different with the Mahalanobis task mapper: while GMMC clusters are learned separately at each teacher for each task as the task is learned, the Mahalanobis classifier is trained by students after sharing, using 5 images/class shared among agents. (6) Agents are implemented in pyTorch and run on desktop-grade GPUs (e.g., nVidia 3090, nVidia 1080).

## 6 Experiments and results

Each LLL agent in our approach is a sequential lifelong learner, capable of learning several tasks in its physical region, one after the other. Hence, before we show full results on the SKILL challenge, we first

compare how well LLL can learn multiple tasks sequentially in a single agent, compared to baselines LL algorithms. This is the standard LL scenario where tasks are learned one after the other and data from previous tasks is not available while learning new tasks.

**Baselines:** We implemented 8 baselines from the literature. For those that require a task oracle, we (unfairly to us) grant them a perfect task oracle (while our approach uses imperfect GMMC or Mahalanobis task mappers). When possible, we re-implement the baselines to use the same pretrained xception backbone as our approach. This ensures a fair comparison where every approach is granted the same amount of pre-training knowledge and the same feature processing ability. The two exceptions are PSP Cheung et al. (2019) that uses ResNet-18, and SUPSUP Wortsman et al. (2020) that uses ResNet-50.

Our baselines fall in the following 3 categories (De Lange et al., 2021): *(1) Regularization methods* add an auxiliary loss term to the primary task objective to constraint weight updates. The extra loss can be a penalty on the parameters (EWC (Kirkpatrick et al., 2017), MAS (Aljundi et al., 2018) and SI (Zenke et al., 2017)) or on the feature-space (FDR (Benjamin et al., 2018)), such as using Knowledge Distillation (DMC (Zhang et al., 2020)). We use EWC as the representative of this category: one agent learns all 102 tasks in sequence, using EWC machinery to constrain the weights when a new task is learned, to attempt to not destroy performance on previously learned tasks. We also use SI, MAS, LwF, and Online-EWC as baselines of this type. *(2) Parameter-Isolation methods* assign a fixed set of model parameters to a task and avoid over-writing them when new tasks are learned (SUPSUP (Wortsman et al., 2020), PSP (Cheung et al., 2019)). We use PSP as the representative of this category: one agent learns all 102 tasks in sequence, generating a new PSP key for each task. The keys help segregate the tasks within the network in an attempt to minimize interference. We used the original PSP implementation, which uses a different backbone than ours. PSP accuracy overall hence may be lower because of this, and thus we focus on trends (decline in accuracy as more tasks are added) as opposed to only absolute accuracy figures. We also used SUPSUP as baseline of this type. *(3) Rehearsal methods* use a buffer containing sampled training data from previous tasks, as an auxiliary to a new task's training set. The buffer can be used either at the end of the task training (iCaRL, ER (Rebuffi et al., 2017b; Robins, 1995)) or during training (GSS, AGEM, AGEM-R, GSS, DER, DERPP (Lopez-Paz & Ranzato, 2017; Chaudhry et al., 2018; Aljundi et al., 2019; Buzzega et al., 2020)). We use ER and as the representative of this category: One agent learns all 102 tasks in sequence. After learning each task, it keeps a memory buffer with 10 images/class (size hence keeps increasing when new tasks are learned) that will later be used to rehearse old tasks. When learning a new task, the agent learns from all the data for that task, plus rehearses old tasks using the memory buffer.

**Accuracy on first task:** To gauge how well our approach is achieving lifelong learning, we plot the accuracy on the first task as we learn from 1 to 102 tasks, in Fig. 4. There is nothing special in our dataset about the first task, except that it is the first one. A good LL system is expected to maintain its accuracy on task 1 even as more subsequent tasks are learned; conversely, catastrophic interference across tasks would rapidly decrease task 1 accuracy with more learned tasks. Overall, our approach maintains the highest accuracy on task 1 over time, and virtually all of the accuracy degradation over time is due to increasing confusion in the task mapper (e.g., curves for Mahalanobis task mapper alone and LLL w/BB w/MAHA are nearly shifted versions of each other). Indeed, once the task is guessed correctly, the corresponding head always performs exactly the same, no matter how many tasks have been learned.

**Normalized accuracy on first 10 tasks:** We compare our method to the baselines on the first 10 tasks, when up to 20 subsequent tasks are learned. A good LL system should be able to maintain accuracy on the first 10 tasks, while at the same time learning new tasks. Because in SKILL-102 different tasks have different levels of difficulty, we normalize accuracy here to focus on degradation with an increasing number of new tasks. For example, the accuracy of our method (LLL w/o BB) when learning a single task is 92.02% for task 1, but only 52.64% for task 6, which is much harder. Here, we define a **normalized accuracy** as the accuracy divided by the initial accuracy just after a given task was learned (which is also the best ever accuracy obtained for that task). This way, normalized accuracy starts at 100% for all tasks. If it remains near 100% as subsequent tasks are learned, then the approach is doing a good job at minimizing interference across tasks. Conversely, a rapidly dropping normalized accuracy with an increasing number of subsequent tasks learned indicates that catastrophic interference is happening.

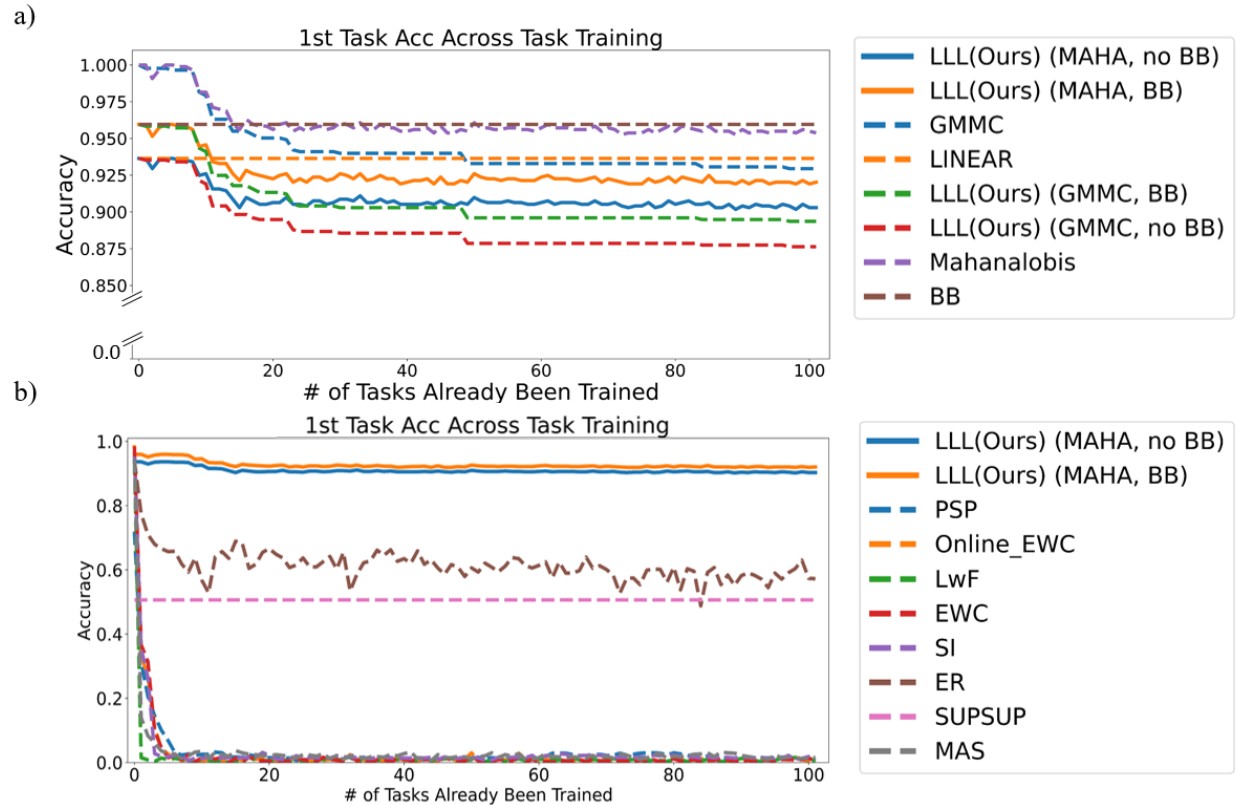

Figure 4: Accuracy on task 1 (learning to classify 102 types of flowers) as a function of the number of tasks learned. a) Comparison between our methods. b) Comparison between our best and other baselines. Our approach is able to maintain accuracy on task 1 much better than the baselines as more and more tasks are learned: while our approach does suffer some interference, task 1 accuracy remains to within 90% of its initial best even after learning 101 new tasks (for the 4 LLL variants, BB=beneficial biases, MAHA=Mahalanobis Distance task mapper, GMMC=GMMC task mapper). In contrast, the accuracy of EWC, PSP, and several other baselines on task 1 catastrophically degrades to nearly zero after learning just 10 new tasks, even though we granted these methods a perfect task oracle. The best performing baseline, ER, is of the episodic buffer type (a fraction of the training set of each task is retained for later rehearsing while learning new tasks), with an un-bounded buffer that grows by 10 images/class. This methods does incur higher (and increasing) training costs because of the rehearsing (Suppl. Sec. D.) Note how SUPSUP does not experience any degradation on task 1, which is a desirable feature of this approach. However, a drawback is that SUPSUP is not able, even from the beginning, to learn task 1 as well as other methods (50.64% accuracy vs. over 90% for most other approaches). We attribute this to SUPSUP's limited expressivity and capacity to learn using masks over a random backbone, especially for tasks with many classes. Indeed, SUPSUP can perform very well on some other tasks, usually with a smaller number of classes (e.g., 91.93% correct on SVHN, 93.18% on Brazillian Coins, 99.11% on UMNIST Face Dataset; see Supp. Fig. S6).

Our results in Fig. 5 show that, although not perfect, our approach largely surpasses the baselines in its ability to maintain the accuracy of previously learned tasks, with the exception of SUPSUP, which suffers no degradation (see caption of Fig. 5 for why).

**Task mapper accuracy after learning 1 to 102 tasks:** To investigate how well our approach is expected to scale with more tasks, we computed task mapper accuracy on all tasks learned so far, after learning 1, 2, 3, ... 102 tasks. This allows us to evaluate degradation with more tasks that is due to increasing confusion in the task mapper, as opposed to being due to classification difficulty of newly added tasks. Results are shown in Fig. 6: Task mapping accuracy starts at 100% after learning 1 task (all test samples are correctly assigned

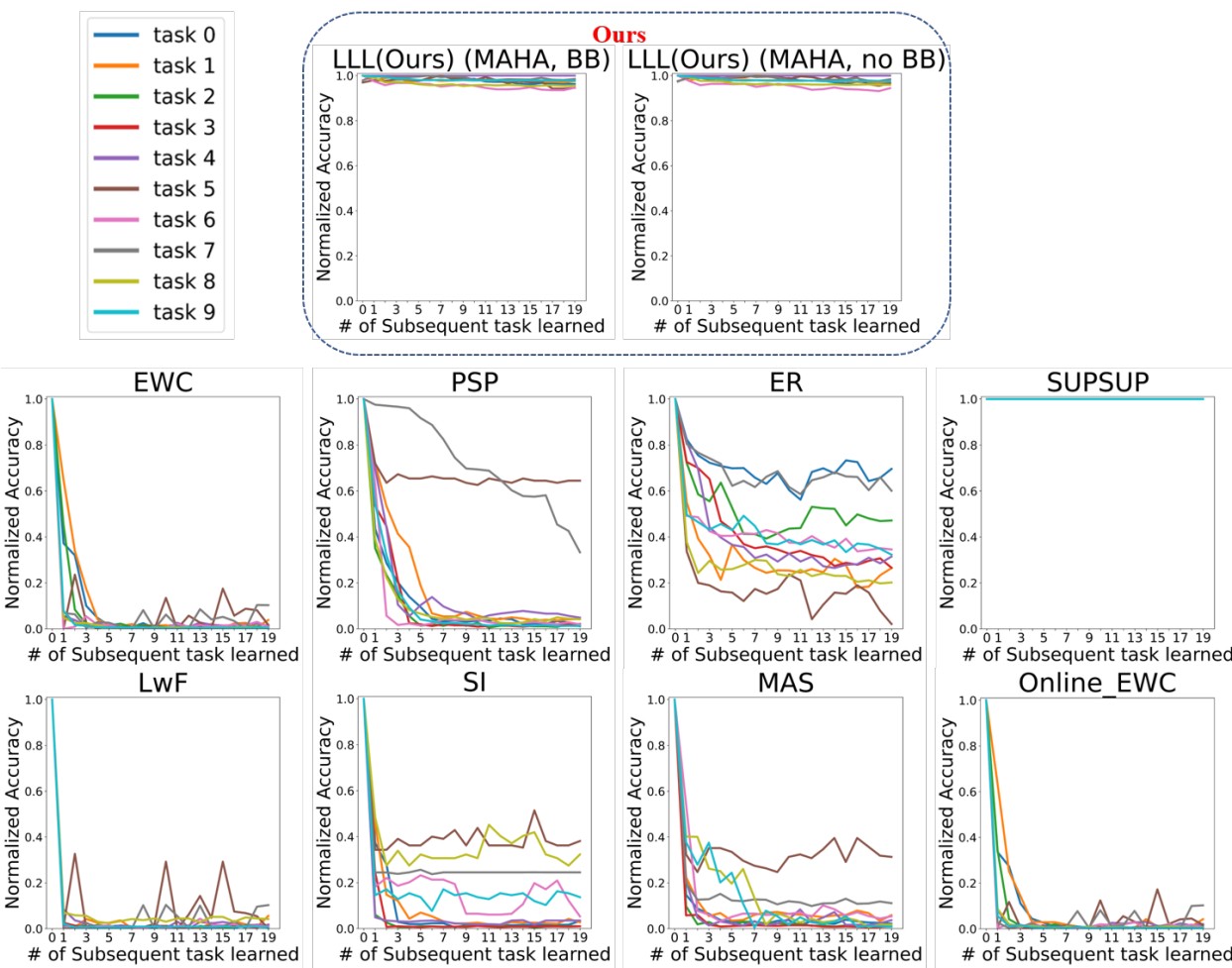

Figure 5: Normalized accuracy on the first 10 tasks (one per curve color) as up to 20 additional tasks are learned. Our LLL approach is able to maintain high normalized accuracy on the first 10 tasks, while all other baselines except SUPSUP suffer much stronger catastrophic interference. SUPSUP is a special case as there is no interference among successive tasks when a perfect task oracle is available. Hence normalized accuracy for all tasks remains at 100%. However, we will see below that the absolute accuracy of SUPSUP is not as good.

to that task by Mahalanobis or GMMC), then decreases as more tasks are learned, eventually still achieving 87.1% correct after 102 tasks for MAHA, and 84.94% correct for GMMC. It is important to note that in our approach, any loss in accuracy with more tasks only comes from the task mapper: once the correct head is selected for a test sample, the accuracy of that head remains the same no matter how many heads have been added to the system. In contrast, other baseline methods may suffer from catastrophic forgetting for both the task mapper and the classification model when more tasks are learned, as further examined below.

When using GMMC task mapping, the regression line is $y = -0.0012x + 0.952$, which intercepts zero for $T = 800$ tasks. Thus, with the distribution of tasks in our dataset, we extrapolate that $T = 500$ is realistic as is. Since task interference in our system only comes from GMMC, pushing beyond $T = 500$ might require more than $k = 25$ GMMC clusters per task, which would increase CPU and communications expenditure. When using Mahalanobis task mapping, the results are similar with an intercept at $T = 978$, though this approach incurs a slightly higher communications cost (discussed below).

**Absolute accuracy:** The normalized accuracy figures reported so far were designed to factor out variations in individual task difficulty, so as to focus on degradation due to interference among tasks. However, they

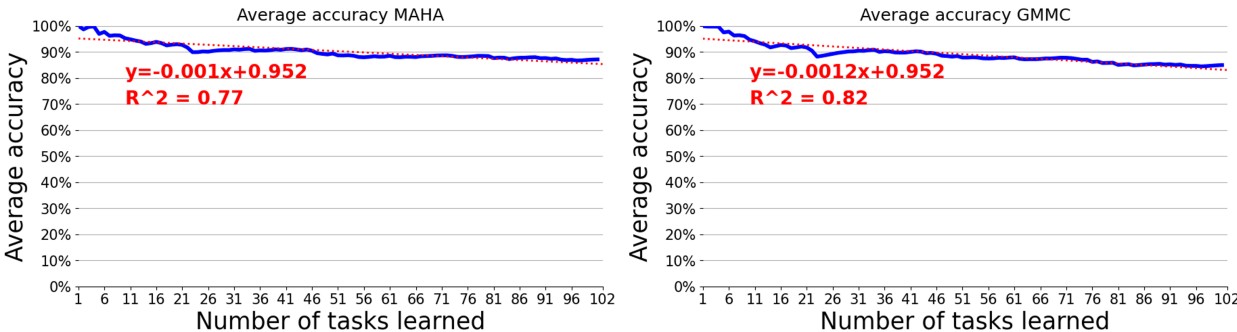

Figure 6: Task mapper accuracy on all tasks learned so far, as a function of the number of tasks learned, when using Mahalanobis (left) or GMMC (right) task mappers. Our approach is able to maintain good task mapping accuracy as the number of tasks increases.

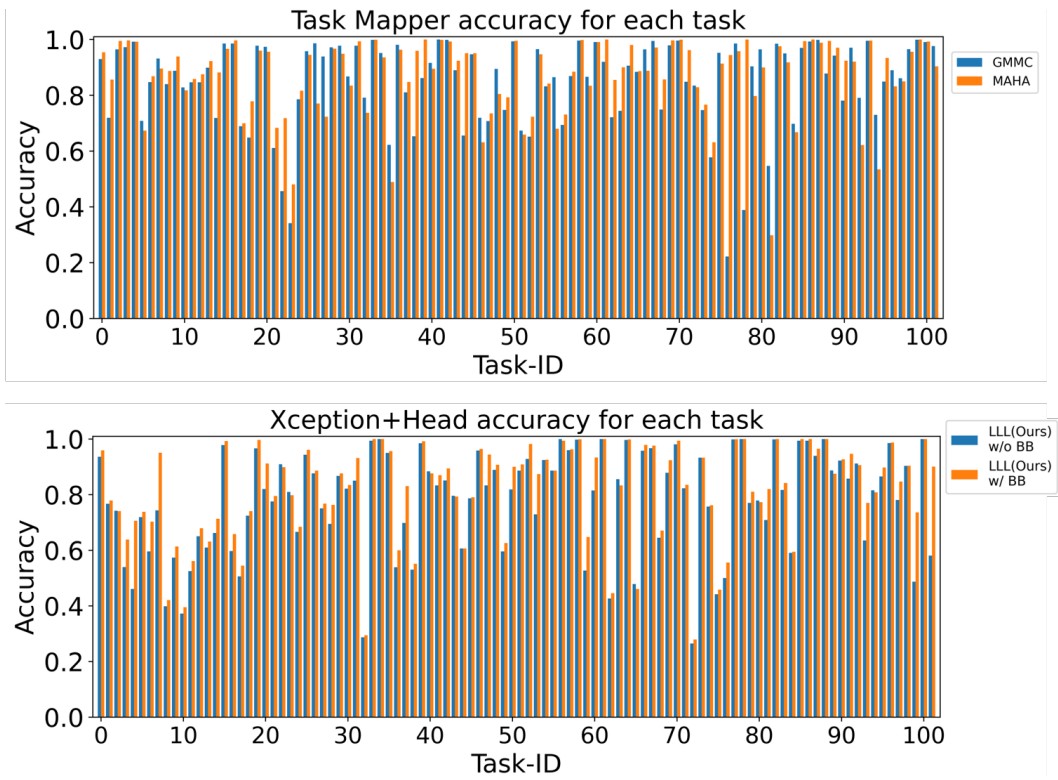

Figure 7: Absolute accuracy per task after learning 102 tasks. (Top) Absolute accuracy of the GMMC and Mahalanobis task mappers alone shows quite a bit of variability, indicating various degrees of overlap among tasks. (Bottom) Absolute accuracy of the main xception+head network alone (with or without BB, assuming perfect task mapper) also shows significant variability, indicating various degrees of difficulty per task. The accuracy with BB is overall slightly higher than without BB (orange bars higher than corresponding blue bars in the bottom panel), as further explored in the next figure.

also factor out the potential benefits of BB in raising absolute task accuracy, and they obfuscate the absolute performance of baselines. Hence, we here also study absolute task accuracy.

We first plot the absolute accuracy of our LLL approach, separately for each task, in Fig. 7, separating whether BB is used or not, and which task mapper is used. This shows that our SKILL-102 dataset provides a range of difficulty levels for the various tasks, and is quite hard overall. BB improves accuracy on nearly

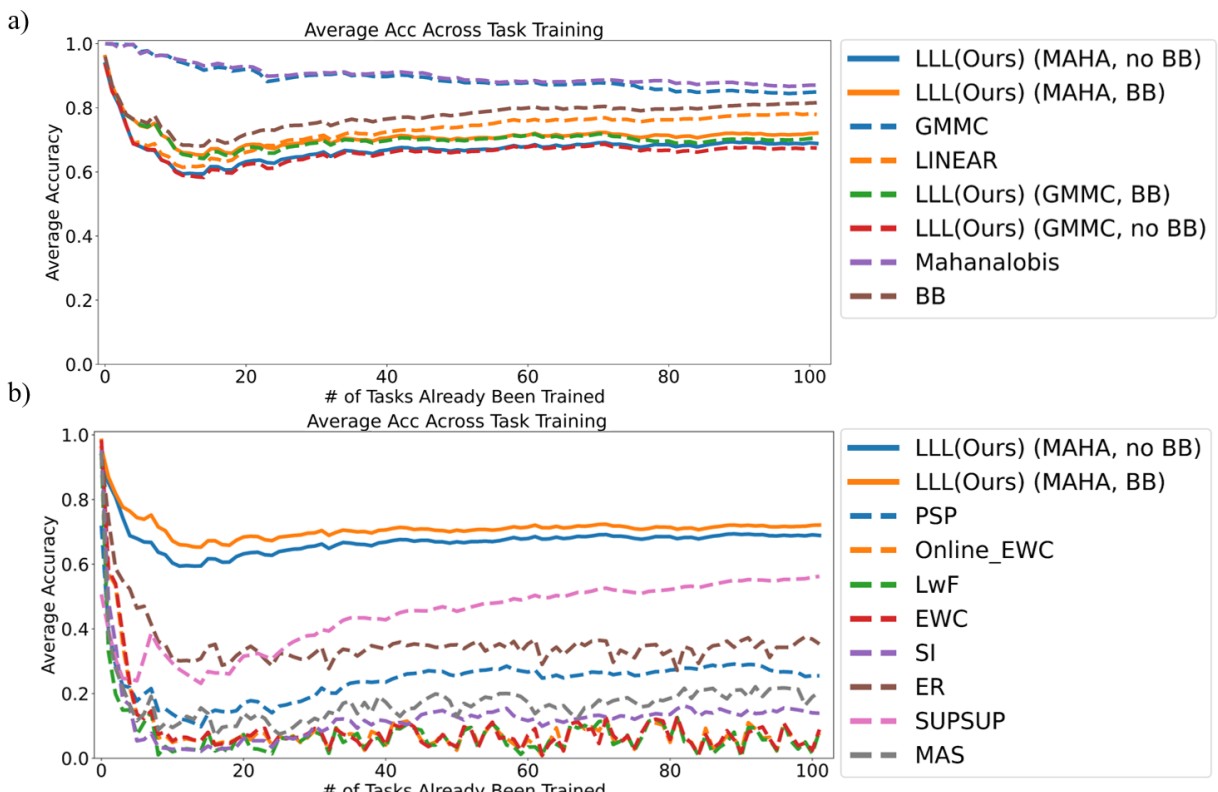

Figure 8: Average absolute accuracy on all tasks learned so far, as a function of the number of tasks learned. Our LLL approach is able to maintain higher average accuracy than all baselines. BB provides a small but reliable performance boost (LLL w/BB vs. LLL w/o BB). The sharp decrease in early tasks carries no special meaning except for the fact that tasks 4,8,10 are significantly harder than the other tasks in the 0-10 range, given the particular numbering of tasks in SKILL-102. Note how again SUPSUP has a low accuracy for the very first task. This is because of the nature of its design; indeed, SUPSUP is able to learn some other tasks in our sequence with high accuracy (Suppl. Fig. S5).

all datasets, at an extra computation cost, detailed below. As promised, BB improves accuracy quite dramatically on some datasets which have a large domain gap compared to ImageNet used to pretrain the backbone (e.g., 31.98 percent point improvement with BB on deepvp that contains dashcam images, 24.92 percent point improvement on CLEVR, 24.5 percent point improvement on Aircraft, 20.75 percent point improvement on SVHN; full details in Suppl. Fig. S5).

We then plot the absolute accuracy averaged over all tasks learned so far in Fig. 8. The absolute accuracy for GMMC and Mahalanobis is the same as before. However, now the absolute accuracies for the full LLL models and for the baselines conflate two components: 1) how much interference exists among tasks and 2) the absolute difficulty of each of the tasks learned so far.

**Computation and communication costs, SKILL metrics:** The baselines are sequential in nature, so trying to implement them using multiple agents does not make sense as it would only add communication costs but not alleviate the sequential nature of these LL approaches. For example, for the EWC baseline, one could learn task 1 on agent A then communicate the whole xception weights to agent B (22.9 M parameters = 91.6 MBytes) plus the diagonal of the Fisher Information matrix (another 22.9 M parameters), then agent B would learn task 2 and communicate its resulting weights and Fisher matrix to agent C, etc. Agent B cannot start learning task 2 before it has received the trained weights and Fisher matrix from agent A because EWC does not provide a mechanism to consolidate across agents. Thus, we first consider one agent that learns all 102 tasks sequentially, with no communication costs.

Table 1: Analysis of computation expenditures and accuracy for our approach and the baselines, to learn all 102 tasks (with a total of 5,033 classes, 2,041,225 training images) in a single agent. Here we select LLL, no BB, MAHA as reference (1x CPU usage) since it is the fastest approach, yet still has higher accuracy than all baselines. For our approach, MAHA leads to slightly higher accuracy than GMMC, at roughly the same computation cost. All baselines perform worse that our approach, even though they also requires more computation than our approaches that do not use BB. BB adds significantly to our computation cost, but also leads to the best accuracy when used with MAHA.

| | Training (MACs) | CPU usage VS. Ours, no BB, MAHA | Average accuracy after learning 102 tasks |
|---|---|---|---|
| LLL(Ours)-Single Agent, no BB, GMMC | 1.73E+16 | ~1x | 67.43% |
| LLL(Ours)-Single Agent, BB, GMMC | 1.56E+18 | ~90.7x | 70.58% |
| **LLL(Ours)-Single Agent, no BB, Mahalanobis** | **1.73E+16** | **1x (reference)** | **68.87%** |
| LLL(Ours)-Single Agent, BB, Mahalanobis | 1.56E+18 | ~90.7x | 72.1% |
| EWC | 1.75E+18 | ~101.3x | 8.86% |
| PSP | 6.28E+17 | ~36.4x | 25.49% |
| ER | 4.53E+18 | ~262.8x | 35.32% |
| SUPSUP | 1.01E+18 | ~58.6x | 56.22% |
| EWC-ONLINE | 1.55E+18 | ~90.1x | 7.77% |
| LwF | 1.56E+18 | ~90.5x | 8.41% |
| SI | 2.07E+18 | ~120.1x | 13.89% |
| MAS | 2.06E+18 | ~119.6x | 20.54% |

Table. 1 shows the computation expenditures (training time in terms of the number of multiply-accumulate (MAC) operations needed to learn all 102 datasets) for our approach and the baselines. Our approach overall has by far the lowest computation burden when BB is not used, yet all 4 variants of our approach perform better than all baselines. BB increases accuracy but at a significant computation cost: This is because, to compute BB biases, one needs to compute gradients through the entire frozen backbone, even though those gradients will only be used to update biases while the weights remain frozen in the backbone.

Our approach presents the advantage that it can also be parallelized over multiple agents that each learn their own tasks in their own physical region. All agents then learn their assigned tasks in parallel. Each agent is the "teacher" for its assigned tasks, and "student" for the other tasks. Then all agents broadcast their shared knowledge to all other agents. As they receive shared knowledge, the students just accumulate it in banks, and update their task mapper. After sharing, all agents know all tasks (and are all identical). As mentioned above, the main source of performance degradation in our approach is in the task mapper, which gets increasingly confused at $T$ increases.

For our baselines, we are not aware of a way to parallelize their operation, except that we were able to create a modified version of SUPSUP that works on several parallel processors. In our modified SUPSUP, each agent learns a mask for each of its tasks, then communicates its masks to all other agents. At test time, we (unfairly to us) grant it a perfect task oracle, as our GPUs did not have enough memory to use the built-in task mapping approach of SUPSUP, given our 102 tasks and 5,033 classes (this would theoretically require 1.02 TB of GPU memory).

Table. 2 shows the computation and networking expenditures for our approach and our modified SUPSUP to learn all tasks in the SKILL-102 dataset. Because some algorithms run on GPU (e.g., xception backbone) but others on CPU (e.g., GMMC training), and because our tasks use datasets of different sizes, we measure everything in terms of MACs (multiply-accumulate operations, which are implemented as one atomic instruction on most hardware). To measure MACs for each component of our framework, we used a combination of empirically measured, framework-provided (e.g., pyTorch can compute MACs from the specification of all layers in a network), or sniffed (installing a hook in some algorithm that increments a counter each time a MAC is executed). To translate communication costs to MACs, we assume a nominal cost of $\alpha = 1,000$ MACs to transmit one byte of data. This is a hyperparameter in our results that can be changed based on deployment characteristics (e.g., wireless vs. wired network). The amount of data shared per task for

Table 2: Analysis of computation and network expenditures for our parallelized LLL approach and our parallelized SUPSUP, to learn all $T = 102$ tasks. Our approach supports any number of agents $N$ such that $1 \leq N \leq T$. Maximum speedup is expected when $N = T$ and each agent learns one task, then shares with all others. Here, we report numbers for $T = 102$, $N = 51$, and each agent learns 2 tasks in sequence. Note that in our approach, accuracy is not affected by $N$, only the amount of parallelization speedup increases with $N$. Note how in this table we still report MACs but taking parallelization into account (e.g., teacher CPU for $N$ agents is single-agent CPU divided by $N$). **Teacher CPU:** Time to learn tasks from their training datasets, plus to possibly prepare data for sharing (e.g., compute GMMC clusters). **Communications:** Our LLL agents communicate either GMMC clusters or Mahalanobis training images, while our modified SUPSUP communicates masks. Here we assume that there is a communication bottleneck at the receiver (student): the shared data from 100 tasks needs to be received serially, over a single networking interface for each student. Hence our communication figures are for all the shared data from all other tasks apart from those an agent learned itself. We convert communication costs to equivalent MACs by assuming 1,000 MACs per byte transmitted. BB adds a small extra communication cost, to transmit the biases. **Student CPU:** For GMMC, students do not do any extra work (hence, student CPU is 0); for Mahalanobis, students compute a covariance matrix for all 102 tasks. **Speedup factor:** is just total MACs for single agent divided by total MACs for parallel agents and by $N$. All approaches here achieve near perfect parallelization ($> 0.99N$, where $1.0N$ is perfect). **Accuracy:** In addition to being faster when BB is not used, our LLL variants still all outperform the parallel SUPSUP in accuracy, by a large margin ($> 10\%$).

| | Teacher CPU (MACs) | Communi--cations (bytes) | Student CPU (MACs) | Total (MACs) | Parallelization efficiency (xN) | CPU usage vs. Ours-SKILL, no BB, MAHA | Average accuracy after learning 102 tasks |
|---|---|---|---|---|---|---|---|
| LLL(Ours)-Multiple Agents, no BB, GMMC | 1.69E+14 | 8.22E+07 | 0.00E+00 | 1.69E+14 | 0.99999519 | ~0.96x | 67.43% |
| LLL(Ours)-Multiple Agents, BB, GMMC | 1.53E+16 | 1.03E+08 | 0.00E+00 | 1.53E+16 | 0.999999934 | ~87.2x | 70.58% |
| **LLL(Ours)-Multiple Agents, no BB, Mahalanobis** | **1.69E+14** | **6.72E+09** | **5.00E+09** | **1.76E+14** | **0.996630551** | **1x (reference)** | **68.87%** |
| LLL(Ours)-Multiple Agents, BB, Mahalanobis | 1.53E+16 | 6.74E+09 | 5.00E+09 | 1.53E+16 | 0.999962712 | ~87.3x | 72.1% |
| Parallel SUPSUP, Perfect Task Oracle | 9.91E+15 | 3.03E+08 | 0.00E+00 | 9.91E+15 | 0.999999697 | ~56.4x | 56.22% |

our approach is quite small (details in Suppl. Sec. G): 813 KBytes/task for LLL with GMMC, no BB; 883 KBytes/task for LLL with GMMC, BB; 1.74 MBytes for LLL with MAHA, no BB; and 1.81 MBytes for LLL with MAHA, BB (on average, given that our tasks have 49.34 classes each on average; see Suppl. Fig. S5).

Our results in Table. 2 show:

1. **Our approach has very low parallelization overhead**, which leads to almost perfect speedup $> 0.99N$ for all variants. Indeed, teachers just learn their task normally, plus a small overhead to train GMMC on their own tasks, when GMMC is used. Communications are less than 2 MBytes per task (Suppl. Sec. G). Students either do nothing (just accumulate received knowledge in a bank) or update their Mahalanobis task mapper.

2. **The baselines have comparatively much higher training cost, yet their performance is poor.** Performance of episodic buffer / rehearsing methods might be improved further by increasing buffer size, but note that in the limit (keeping all training data for future rehearsing), this gives rise to a $> 5,000\times$ increase in training time (Suppl. Sec. D).

# 7 Shared Knowledge Accumulation, Reuse and Boost

As our system learns many tasks, it may occur that some tasks overlap with others, i.e., they may share similar images and/or class labels. Here, we explore two approaches to handle such overlap.

## 7.1 Corrective approach to task overlap/synergy

Since our LLL learners can learn a large number of tasks while solving SKILL problems, some synergy can occur across tasks if one is able to detect when two classes from two different tasks are equivalent, as shown in Fig. 9. We implemented a method to compare the semantic distance of the predicted class name and the actual class name. Originally, after the GMMC infers the task that a test image may come from, we would immediately consider the image as misclassified if the predicted task is wrong. With the consideration of semantic similarity between class names, we will now always load the prediction head corresponding to the predicted task given by GMMC and use it to infer the class name. If the class was guessed incorrectly, but, in the end, the final class name is equivalent to the correct one, then we can declare success on that particular test image (Fig. 9). To obtain the pairwise similarity, we constructed a similarity matrix that stores the semantic distance, measured by the cosine similarity of word embeddings, for all the class names. Those embeddings were obtained from CLIP's (Radford et al., 2021) text encoder based on GPT-2. If the similarity between the predicted class name and the actual class name is greater than a threshold (empirically chosen for now), then we declare it a correct prediction.

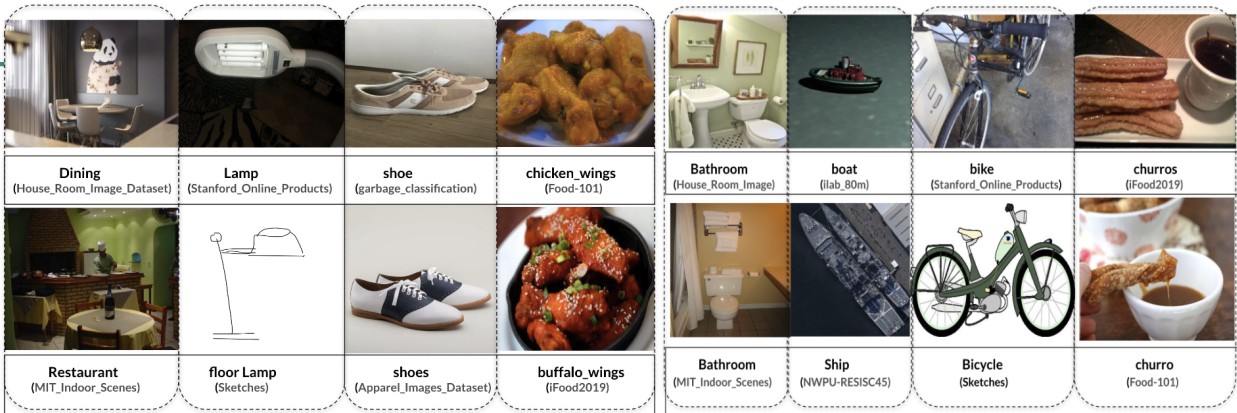

Figure 9: Left: similar classes with a cosine similarity in the CLIP embedding greater than 0.90. Right: similar classes with a cosine similarity greater than 0.95. This can help correct spurious errors where, for example, a test image from class "bike" from the Stanford_Online_Products dataset could also be considered correctly classified if the system output were "bicycle" from the Sketches dataset.

As our 102-task dataset contains 5,033 object classes, the full similarity matrix is triangular 5,033 x 5,033 (too large to display here).

The approach yields a small but consistent improvement in accuracy (Fig. 10). This is one way in which we can handle possible overlap between tasks, which may inevitably arise when large numbers of tasks are learned.

## 7.2 Learning approach to task overlap/synergy

The ability to reuse learned knowledge from old tasks to boost the learning speed and accuracy of new tasks is a potential desirable feature of LL algorithms. Here, this might be achieved if, when learning a new task, an LLL agent could "borrow" the knowledge of old tasks, from not only itself but also the shared knowledge from any other agents.

One important design feature of our LLL agents is that they can share *partial heads* across tasks: Our heads are a single layer with 2,048 inputs (from the xception backbone) and $c$ outputs for a task with $c$ classes. Thus, each of the $c$ output neurons is connected to the 2,048 inputs, but there are no lateral connections. This means that we can consider each of the $c$ output neurons individually as evidence provider for their associated class (remember the analogy to "grandmother cells" in Introduction). We can then cherry-pick sets of 2,048 weights corresponding to individual classes of previously learned tasks, and use them as initialization for some similar classes in new tasks to be learned. As we show below, this greatly accelerates learning of the

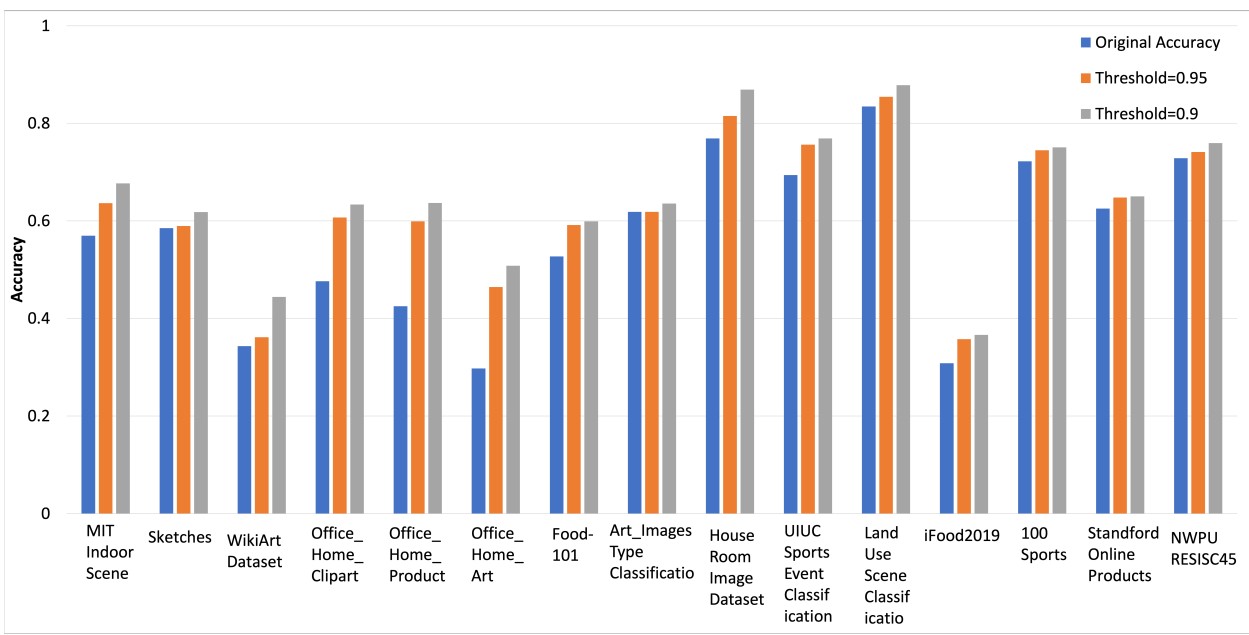

Figure 10: Correcting spurious errors by realizing when two distinct classes from two tasks actually are the same thing. The approach provides a small but consistent improvement in accuracy over baseline (which declares failure as soon as task mapping failed), here shown on 15 datasets that have some overlap.

similar new classes, compared to starting with randomized initial weights, and also yields higher accuracy on these new classes.

**1) New task is a combination of old tasks:** To validate our idea, a new learning paradigm is proposed to use previously learned weights when a new task contains classes that were already learned previously. This experiment considers two datasets and two sets of weights representing the old knowledge, and a new dataset that contains all classes from both datasets. Simply normalizing and concatenating the linear weights leads to poor performance. Hence, instead, we normalize the weights during training by their *p-norm*, and concatenate the normalized weights as the new task's weights. The experiment was conducted over 190 combinations of 2 datasets chosen from 20 datasets, and the average results show that there is a very small accuracy loss initially (epoch 0). After a few extra training epochs, we reach a higher accuracy than training new weights from scratch (random initialization; Fig. 11).

**The mathematical version.** During training, a constraint is added. Let $W_1$ be the full set of $2,048 \times c$ weights of the first dataset, and $W_2$ be the weights of the second dataset, where $W_1 = \begin{pmatrix} ...w_1^1... \\ ... \\ ...w_n^1... \end{pmatrix}$ and $W_2 = \begin{pmatrix} ...w_1^2... \\ ... \\ ...w_n^2... \end{pmatrix}$ . Normal Linear Layer training's forward path is $\hat{y} = \text{W}x$. Define W' as $\begin{pmatrix} w_1' \\ ... \\ w_n' \end{pmatrix}$ where $w_i' = w_i/\text{p-norm}(w_i)$ Hence, during training, Linear Layer training's forward path is $\hat{y} = \text{W}'x$. And concatenate($W_1$', $W_2$') was used as the weights for the new combined task.

Since the weights are normalized class-wise and not task-wise, this method can be used on any combination of previously leaned classes. For example, for 10 tasks containing 100 classes $c_1$ $c_{100}$ and a new task containing $c_1, c_3, c_{10}, c_{20}$, we can simply find the corresponding $w_1', w_3', w_{10}', w_{20}'$, and concatenate them together.

**Choice of $p$:** We find that using the 2-norm causes the classifier to converge to a state where all weight vectors have the same magnitude, which causes an accuracy drop for the old task. Hence, we choose to use the infinity norm, which is still modulated by the weight magnitudes, and is still easy to transfer.

**2) New task is different but similar to old tasks.** In the previous setting, we assumed the new task classes are a combination of different old task classes. In a more general situation, the new task classes are

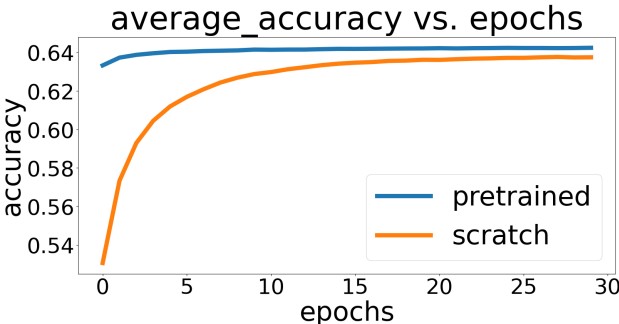

Figure 11: Learning speed for a given object class when the corresponding weights are initialized randomly (orange) vs. from previously learned weights of a similar object class found in one of the previously learned tasks (blue), averaged for 190 combinations of two previously learned tasks. In this example, best accuracy is already reached after just 1 to 2 training epochs in the blue curve. In contrast, it takes up to 30 epochs to train from random initialization, with still a final accuracy in the orange curve that is lower than the blue curve. This approach hence leads to significant learning speedup when tasks contain some similar classes.

all new classes that all other agents never learned before, but we could still borrow some learned knowledge from similar learned classes. For instance, as shown in Fig. 9, the knowledge of classes shown on top of the figure may be helpful to learn the new classes shown at the bottom.

We conduct four different experiments (for 4 pairs of datasets that share some related classes) to show the knowledge boost when we learn a new task. We first check if a learned old task shares similar knowledge with the new one. For instance, before we learn the *MIT indoor scenes dataset*, we find that the *House Room Image Dataset* contains classes that are similar to the new classes, in the CLIP embedding space. So we match each class from *MIT indoor scenes dataset* to the previously learned classes, which in this case come from the *House Room Image Dataset*. If the class similarity is larger than a threshold, we treat it as a matched class, then we use the similar old class weights to initialize the weights of the new class. If a new class was not matched with old classes, we use random initialization. We also conduct a corresponding controlled experiment by using random initialization for all new classes. The results of all 4 experiments are shown in Table 3

|  | Datasets | initialization | All | 10 shot | 5 shot | 3 shot |
|---|---|---|---|---|---|---|
| Old task | House Room Image Dataset | random | 0.86 | 0.77 | 0.73 | 0.52 |
| New task | MIT Indoor Scenes | ours | 0.89 | 0.83 | 0.8 | 0.71 |
| Old task | Standford Online Products | random | 0.78 | 0.61 | 0.61 | 0.6 |
| New task | Office Home Product | ours | 0.8 | 0.62 | 0.59 | 0.6 |
| Old task | 100 Sports | random | 1 | 0.98 | 0.92 | 0.92 |
| New task | UIUC Sports Event Dataset | ours | 1 | 0.99 | 0.97 | 0.97 |
| Old task | iFood2019 | random | 0.61 | 0.42 | 0.35 | 0.3 |
| New task | Food-101 | ours | 0.64 | 0.5 | 0.46 | 0.43 |

Table 3: Boosted LLL learning when previously learned weights from similar classes can be used to initialize learning of new classes. We repeat the experiment with either learning from all images in the training set of the new task, or only 10, 5, or 3 images per class. Overall, re-using previously learned weights of similar classes boosts accuracy, usually (but not always) more so when the new task is only learned from a few exemplars (which is much faster than learning from scratch from the whole dataset).

In a more general learning scenario, the new task classes may correspond to similar but not necessarily identical classes in different old tasks. For example, having learned about SUVs may help learn about vans more quickly. Here we conduct two new experiments (Fig. 12). In EXP-1, the new task is sketch image classification, and the classification weights of each class are initialized from a learned old class that

is harvested from different learned tasks with the help of CLIP-based word matching. For instance, the classification weights of *van* are initialized with the weights of *SUV* from the previously learned *Stanford-CARs dataset*; the classification weights of *Topwear* are initialized with the weights of *t-shirt* from the learned *Fashion-Product* dataset, etc (total: 5 pairs of classes). The results show that our initialization leads to better performance (0.98 v.s. 0.93) and faster convergence (3 v.s. 4 epochs) compared with random initialization. This shows that our method can reuse the shared knowledge from other agents to improve the learning of new tasks. Similar performances are shown in EXP-2, which is identical to EXP-1 except that it uses 5 different pairs of classes.

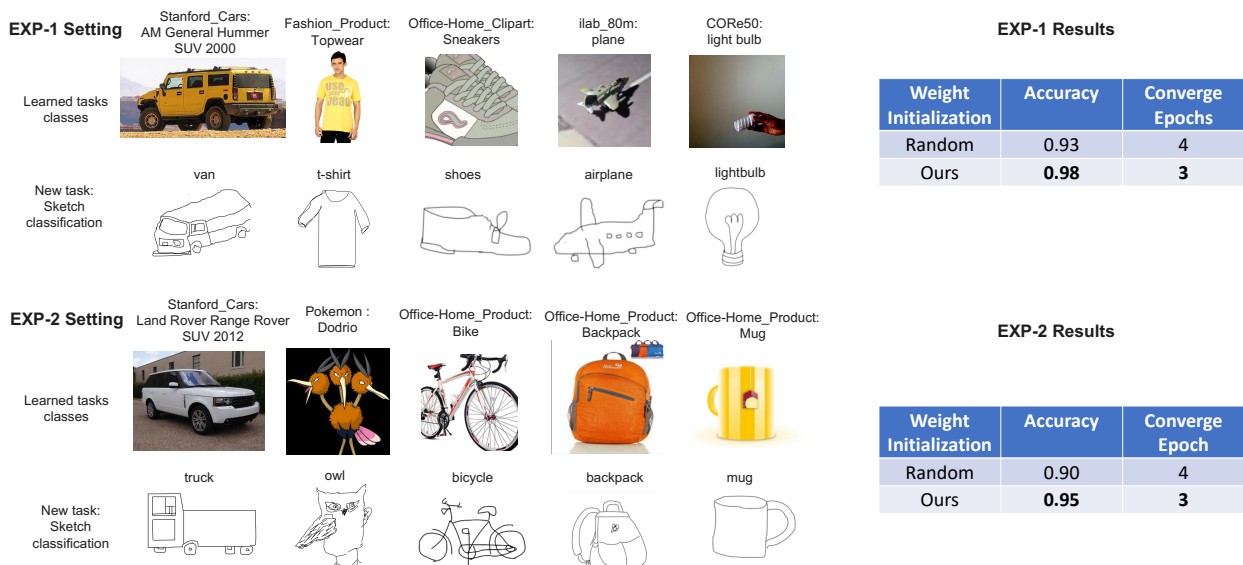

Figure 12: Two experiments where the weights from previously learned similar but not identical classes are successful in boosting learning of new classes. Left: pairs of similar classes (according to CLIP). Right: accuracy achieved with weight transfer vs. random initialization.

## 7.3 Further boost with Head2Toe

A possibly complementary approach to BB to address domain gaps is Head2Toe (Evci et al., 2022), where the last layer can now directly draw from potentially any of the previous layers in the backbone. This has been shown to help alleviate domain gaps, as some of the lower-level features in the backbone may be useful to solve tasks with a big gap, even though the top-level backbone features may not. However, Head2Toe has a very high computation cost to select which layers should connect to the head, which is why we have not used it in our main results. Here, we explore how that cost of selection of the most appropriate layers to connect to the head for a given task can be eliminated by re-using the computations already expended for BB: Intuitively, layers which have large absolute BB magnitude may also be the most useful to connect to the head.

Compared to the conventional Head2Toe (Evci et al., 2022) with two-stage training (first, select which layers will connect to the head, then train those connections), our new BB+H2T uses the biases that have been previously trained and stored in the BB network for feature selection. Specifically, we first concatenated all the biases in the BB network and selected the top 1% largest biases. Then, we picked the feature maps corresponding to the selected indices, average pooled them and flattened them to 8,192-dimensional vectors. After that, we concatenated all flattened feature vectors along with the logits of the last layer (after pooling layer, before softmax) in the BB network. Finally, we trained the concatenated vector with Adam optimizer, 0.001 learning rate, and 100 epochs. This approach, when combined with BB and MAHA, improved performance averaged over all tasks by 0.78% (when a perfect task mapper is available; or by 0.56% when using MAHA).

# 8 Discussion and Future Works

We have proposed a new lightweight approach to lifelong learning that outperforms all baselines tested, and also can benefit almost perfectly from parallelization. We tested the approach on a new SKILL-102 benchmark dataset, which we believe is the largest non-synthetized lifelong learning challenge dataset to date. While many previous efforts have employed many tasks, those were usually synthesized, e.g., as permutations over a single base dataset (e.g., 50 permuted-MNIST tasks in Cheung et al. (2019)). SKILL-102 contains novel real data in each task, with large inter-task variance, which is a better representative of realistic test scenarios. Our proposed lightweight LL points to a new direction in LL research, as we find that one can simply use lightweight task-specific weights (head) combined with maximizing the leverage obtained from task-agnostic knowledge that is rapidly adapted by a compact BB module to handle each new task. Our results show how this lightweight design is better able to handle large scale lifelong learning tasks, and also solves our SKILL challenge very well.

We credit our good performance on both sequential lifelong learning and the SKILL challenge to our particular lightweight lifelong learner design: a *fixed backbone*, which represent task-agnostic knowledge shared among agents, to minimize the complexity of task-specific knowledge parameters (the head); *Beneficial Biases*, which on-demand shift the backbone to solve possibly large domain gaps for each new task, with very compact parameters; a *GMMC/MAHA* global task anchor for learned tasks, representing the tasks in the common task-agnostic latent space of all agents, which is easy to share and consolidate, and which eliminates the need for a task oracle at test time. Our results show that combination of this three components help our LLL work well.

Our approach uses a pretrained backbone to represent task-agnostic knowledge, which our results show is a very effective strategy. For fair comparison, we also use the same pretrained backbone as the initialization for the baselines (except PSP and SUPSUP; see above). However, our fixed backbone design often cannot handle large domain gaps between new tasks and ImageNet. This is the reason why we proposed BB to relieve the domain gap by shifting the fixed parameters towards each new task with compact biases. Similar to other parameter-isolation methods, our model structure is growing on demand (though slowly) with the number of tasks (we add a new head per task, while they add new masks, keys, etc). Rehearsal-based baselines (e.g., ER) also grow, by accumulating more rehearsing exemplars over time. While some baselines do not grow (e.g., EWC), they also perform very poorly on our challenging SKILL-102 dataset.

To further speed up our method with BB, we could use BB on only partial layers. For instance, if we use BB only the last half of the layers in the backbone, we will use only half of the current time to train the model. In future experiments, we will test whether this still gives rise to a significant accuracy benefit.

Currently, we use the CLIP embedding space to match a new class with learned old classes, which uses only language knowledge (class labels). As a future work, we will use GMMC as a class matching mechanism to utilize the visual semantic information for matching. Specifically, when a agent learns a new class, the agent will collect a few shots (e.g., 10 images) of the new class and then use the GMMC mapper (trained on all previous tasks) to decide whether these images belong to a learned task or not, with a threshold. If most of the images are matched to a learned task, we can then summon the shared head of that task to classify them, now to obtain a possible match for individual previously learned classes. If most of the images are classified into one previously learned specific class, we can use the weights of that class to initialize the new class weights, similar to what we have done in Sec. 7.

A good task mapper is essential in our approach if one is to forego the use of a task oracle. Thankfully, our results show that task mapping can be achieved with high accuracy even with a simple model like GMMC (over 84% correct for 102 tasks). Indeed, in our SKILL challenge, the task mapper is only solving a 102-way classification problem at the granularity of tasks, vs. a 5,033-way full SKILL classification challenge. Here, we focused on GMMC and MAHA, but many other alternatives could be investigated in future work. Our choice of GMMC was based on previous research that compared it to several other techniques, including self-organizing maps and multilayer perceptrons (Rios & Itti, 2020).

## 9 Conclusions

We have proposed a new framework for shared-knowledge, parallelized LL. On a new, very challenging SKILL-102 dataset, we find that this approach works much better than previously SOTA baselines, and is much faster. Scaling to $> 500$ difficult tasks like the ones in our new SKILL-102 dataset seems achievable with the current implementation.

**Broader impacts statement:** We believe that LLL will spur a new generation of distributed LL systems, as it makes LL more accessible to edge systems and more parallelizable. Thus, broader impacts are expected to be positive in enabling more lightweight devices to learn at the edge and to share what they have learned.

**Acknowledgement:** This work was supported by DARPA (HR00112190134), C-BRIC (one of six centers in JUMP, a Semiconductor Research Corporation (SRC) program sponsored by DARPA), and the Army Research Office (W911NF2020053). The authors affirm that the views expressed herein are solely their own, and do not represent the views of the United States government or any agency thereof.

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
