# OpenReview forum: "Lightweight Learner for Shared Knowledge Lifelong Learning"
_TMLR — Accepted by TMLR_

### Review · Reviewer_vikF · 2023-03-29

**Summary Of Contributions:**

The paper proposes a new class of lifelong learners dubbed "Lightweight Lifelong Learning" (LLL). LLL is based on a pretrained and (mostly) frozen feature extraction backbone (Xception) and learns task-specific heads and updates biases in the backbone. Moreover, a task mapping is learned. LLL learns those task-specific parameters in a distributed manner on multiple agents that communicate learned parameters with each other. LLL is evaluated on the newly proposed SKILL-102 benchmark, where it outperforms other approaches in terms of performance, compute requirements and communication overhead.

**Audience:**

Yes

**Broader Impact Concerns:**

The reviewer does not have broader impact concerns

**Claims And Evidence:**

No

**Requested Changes:**

- Conducting additional experiments on established and publicly available benchmarks would be required in my opinion for acceptance.
- As an additional baseline, it would be helpful to understand how a zero-shot CLIP based classifier would perform on the proposed SKILL-102 benchmark (with queries of the form "an image of a <class>" with the respective class added).
- The SKILL-102 dataset needs to be released publicly, to ensure reproducibility and to be considered a contribution of this paper. Moreover, is it ensured that the pretraining data of Xception (ImageNet1k?) is disjoint from the data contained in SKILL-102?
- It is somewhat surprising that the simple task mappers considered in the paper that operate in image space have relatively high accuracy. It would be good to understand how they can accomplish this and to release code and data to be able to conduct an independent reproduction of the results.
- Figure 4a (and some later ones): if all methods have an accuracy > 85%, the range of the y-axis should not start with 0%, which makes the figure hard to read. Also, when possible, error-bars/confidence intervals would be desirable.
- Minor: In caption of Figure 3, please also mention that BBs fall under the shared "knowledge"
- Minor: The linear regression line in Figure 6 does not make sense, as the trend is clearly non-linear.

- Minor: I assume statements like "31.98% improvement with BB" rather mean "31.98 percent point improvement with BB"?

- Minor: Figure 6 would be more readable as a scatter plot (w BB vs. w/o BB)

- Minor: Figure 2c "other datasets' plots" is not a descriptive Figure caption, please add more details here.


**Strengths And Weaknesses:**

Strengths:
- The setting of "lifelong learning" is a relevant problem for the audience of TMLR (and the broader field)
- New benchmarks like SKILL-102 are valuable for the field
- Figure 1 provides a good overview of the approach
- It is laudable that the authors also evaluate their approach in terms of compute and communication overhead
- The concept of incrementally task mappers is novel (to my knowledge) and works surprisingly well
- The authors conduct extensive experiments on the SKILL-102 benchmark, comparing also to prior work (partly reimplemented)

Weaknesses:
- The considered setting comes with strong assumptions such as the ones listed below. These should be added in the "Assumptions" part in Section 3. The relevancy to the broader audience could be increased by considering a more general setting. Assumptions:
    * all tasks being image classification tasks
    * perfect task oracle at training time
    * no two agents ever face the same task during training
    * clear separation into training and testing phase (unrealistic in lifelong learning)
    * trustworthy, reliable communication between all agents
- The proposed SKILL-102 dataset is derived from another dataset (DCT), a paper on this dataset being currently under review. It is unclear if SKILL-102 can be seen as an independent contribution of this paper as it is a subset of DCT. Moreover, the dataset is not publicly available (at least this is note clear from the paper).
- Evaluation is done essentially on a proprietary in-house benchmark only; this does not allow reproducibility and I don't see these experiments as "convincing evidence of the papers claims" as required by TMLR.
- The presentation of the paper could be improved, overall I don't think 21 pages are required for exposition of the methods and evaluations conducted in this work.
- The number of parameters of LLL grows linearly with the number of tasks. That is a disadvantage compared to some of the prior work
- Section 7 feels unrelated to the core content of the paper and also somewhat unfinished (some parts of the exposition read more like a preliminary blogpost "It is tricky to choose value of p.", "On a small sample of classes tested so far" etc.). I think the paper might be improved by removing this section or moving it to the appendix (alternatively, a major revision of this part is required) . I am also a bit sceptical about the part on "similarity between the predicted class name and the actual class name is greater than a threshold": if the threshold goes to -infinity, then all classes are mapped onto a single class and accuracy goes trivially to 100%, if I understand correctly what the authors are doing. That would clearly be problematic.

---

> ### Author Response · Authors · 2023-04-20
> **Feedback for Reviewer vikF (part 1)**
>
> Dear reviewer,
>
> Thank you for your detailed and very useful comments!
>
> **[Q1 SKILL-102 dataset]** SKILL-102 is an independent contribution of this paper. We will publicize the dataset upon acceptance.
> DCT is an extension of SKILL-102 and contains more details about other applications (beyond lifelong learning and SKILL) that will show as a technical report in the future.
> To avoid confusion of readers, we removed the mention of DCT from the revised paper.
>
> **[Q2 Evaluation and new experiments]** All 102 datasets are public datasets released before. Our SKILL-102 is also a superset of some well-acknowledged datasets:  The first 8 tasks have been used in Aljundi et al, 2018 [1] and Wen at al (2021) [2]; The Fine-Grained 6 Tasks dataset (Fig. 2) is a subset of SKILL-102 except for the first task (which is related to ImageNet). F-CelebA [3] also overlaps with SKILL-102. To the best of our knowledge, SKILL-102 is the most challenging lifelong learning benchmark. So, SKILL-102 is in-house for now, but by no means proprietary. We will distribute it along with this paper and we hope that it will become the new preferred benchmark for future LL research.
>
> We also conducted new experiments on another large-scale, well established, and public benchmark: Visual Domain Decathlon [4] with 10 tasks, which we had mentioned in Fig. 2c. The results are as in the new Suppl. Fig. S8. The conclusions are the same as for SKILL-102: our approach (solid curves) still by far outperforms the baselines (dashed curves).
> Yes, as far as we know, ImagineNet-1k is disjoint from the data in SKILL-102. This was a consideration when we designed SKILL-102 (e.g., available dataset Stanford Dogs is a subset of ImageNet and hence was not included in SKILL-102).
>
> **[Q3 Zero-shot CLIP performance]** Thanks for your suggestion, we conducted the CLIP experiments on the SKILL-102 benchmark dataset following the paper "CLIP model is an Efficient Continual Learner. [5]”. For each image embedding after image encoder, we compute the cosine similarity with all 5,033 candidate classes in 102 tasks and select the class with the largest similarity score as the classification result. The average performance on SKILL-102 is 19.01%, much worse than our LLL performance.
> On some tasks, especially the datasets with common classes and clear labels, the performance is relatively good, for instance: Food 101 (acc: 65.26%), which consists of different food classes; Oregon Wildlife (acc: 69.71%), which consists of different animal species. However, the performance is poor for the tasks that need specific domain knowledge or with no proper class label to describe the classes.
>
> For instance, PatchCamelyon (acc: 0.02%) is a medical histology dataset (slices of tissue viewed in a microscope). Labels are “negative” or “positive” for whether or not there is metastatic tissue in an image. When we prompt CLIP for “an image of a negative”, that prompt does not help decide whether the image contains cancer cells or not. In order for CLIP to be useful, one would first need to examine each dataset, introduce new expert domain knowledge, and re-label the dataset in a way that CLIP could understand (e.g., “a histology slice containing cancerous cells” vs. “a histology slice not containing cancerous cells”). For up to 5,033 classes, this would likely require significant amounts of expert annotation and prompt engineering work. Hence, although appealing, the idea of using CLIP, we find, is far from being a simple endeavor.
>
> Additionally, if more than one task use the same class name, CLIP cannot distinguish which task that image belongs to. For instance, the Sketches and the Stanford Online Product datasets have class names “bicycle” and “chair”. Likewise, the Houseroom Images dataset and MIT Indoor Scene dataset both have “bathroom” and “livingroom", etc. For these overlapping class names, CLIP will return exactly the same score for more than one of the 5,033 classes (e.g., “an image of bathroom” for the bathroom class in Houseroom Images is exactly the same prompt as for the bathroom class of MIT Indoor Scene). Thus the same prompt may be used up to k times, for the same class name used in k different datasets. In our experiments, as long as the real label is in the k labels, we treat CLIP has a correct prediction. However, this is another open challenge to creating a good CLIP baseline in the real world.
>
> Reference
>
> [1] Aljundi, et al. "Memory aware synapses: Learning what (not) to forget." ECCV. 2018.
>
> [2] Wen, et al. "Beneficial perturbation network for designing general adaptive artificial intelligence systems." IEEE TNNLS 33.8 (2021): 3778-3791.
>
> [3] Ke, et al. "Continual learning of a mixed sequence of similar and dissimilar tasks." NeurIPS (2020): 18493-18504.
>
> [4] Rebuffi, et al. "Learning multiple visual domains with residual adapters." NeurIPS 30 (2017).
>
> [5] Thengane, et al. "CLIP model is an Efficient Continual Learner." arXiv:2210.03114 (2022).

---

> > ### Author Response · Authors · 2023-04-20
> > **Feedback for Reviewer vikF (part 2)**
> >
> > **[Q4 Assumptions]** Thanks for your suggestion! We updated the Assumptions section of the paper. Here are the details:
> >
> > *(1) "all tasks being image classification tasks"*:
> >
> >  Yes - this is mentioned in Sec 4
> >
> > *(2) "perfect task oracle at training time"*
> >
> >  Yes, we added this to “Assumptions”
> >
> > *(3) "no two agents ever face the same task during training"*
> >
> >  Actually, our approach does not preclude that several agents could learn the same task. At test time, it will likely be a toss as to which head would be used, but if several agents learned the same task well, any of the corresponding heads should be able to solve that task reasonably well.
> >
> > *(4) "clear separation into training and testing phase"*
> >
> > yes, added to “Assumptions”
> >
> > *(5) "trustworthy, reliable communication between all agents"*
> >
> > Generally yes, but not strictly required. If an agent is disconnected, then others will just not benefit from its knowledge, but they will continue to operate nevertheless. Please see the answer to Q3 of reviewer 4LNS (what happens if half of the agents are destroyed).
> >
> > **[Q5 Task mappers]** The GMMC and Mahalanobis task mappers learn the statistics of each task in a shared latent space (the 2048D embedding provided by our frozen Xception backbone). To address your question, we applied our approach to a completely different dataset sequence: the Visual Domain Decathlon dataset with 10 tasks. Performance was again very good (see new Suppl. Fig S8).
> > We believe one of the reasons for the good performance is the good embedding in which our task mappers operate. The 2048D embedding of Xception provides clearly separable latent representations for our task mappers.
> > We will release all the code and data upon publication for reproduction.
> >
> > **[Q6 Parameter grow]**
> > Yes, the parameter of LLL grows with the number of tasks, though relatively small compared with model backbone: For Xception, which has 22.9M weights with 91 MBytes storage, LLL grows < 2MBytes/task.
> > Some of the baselines (parameter-isolation based and replay-based) also grow with the number of tasks:
> > - PSP stores one task-specific PSP-key for each task [9MBytes/task].
> > - SUPSUP stores one task-specific supermask for each task [3 MBytes/task].
> > - Replay-based ER stores 10 images per class for future rehearsing (with 49 classes/task on average in SKILL-102, and 299x299x3 images, that is 130 MBytes/task).
> >
> > Only the regularization-based baselines (EWC, MAS, and SI) do not have a growing storage requirement; however, their performance is very low in the challenging SKILL-102 dataset.
> >
> > **[Q7 Section 7]** Thanks for your comments, we have refined (and shortened a bit) this part, but we think reusing learned knowledge from old tasks to boost the learning speed and accuracy of new tasks is a potentially desirable feature for our LLL. As reviewer UV1p mentioned, “they would be interesting to the wider community”.
> > For the similarity threshold between the predicted class name and the actual class name, in practice, we will use a relatively high threshold (0.95 or 0.9) as shown in Fig. 12 and Fig. 11.
> >
> > **[Q8 Paper length]** Thanks for your suggestion, we will try to shorten the paper.
> >
> > **[Q9 Figure visualization]** Our initial setting is to keep the y-axis consistency between Fig. 4a and 4b for easier comparison. Thanks for your suggestion. We have updated Figure 4a in the revised main paper by zooming in on the region over 0.85.
> >
> > **[Q10 Minor]** Thanks for your suggestion! Here are more detailed feedback:
> >
> > (1) *"caption of Figure 3"*
> >
> > We have updated the caption of Fig. 3 by adding the BBs. Thanks!
> >
> > (2) *"linear regression line in Figure 6"*
> >
> > Well, if we wanted to use a different model, we would need an underlying theory or reason to justify the extra degrees of freedom. In this case, really there are some fluctuations and deviations from linear mostly because task difficulty varies from one task to another. Thus, this is not a consequence of some more general property of our dataset sequence. Hence, because we would be hard pressed to justify why, e.g., a quadratic model is a better representation of this data, we would like to keep it as simple as possible, i.e., linear.
> >
> > (3)*"31.98 percent point improvement with BB"*
> >
> > Yes, thanks for your clarification! We have modified this in the revised version.
> >
> > (4) *"Figure 6 would be more readable as a scatter plot (w BB vs. w/o BB)"*
> >
> > We believe you meant Fig 7 (the one with the pairs of blue and orange bars). Indeed, a scatter plot is interesting, but we need to preserve the task ID, which is quite important here (e.g., task mapper accuracy is low for task 23, which we can then look up in Suppl. Fig. S5 to try to understand why). This would require tagging every scatter point with a task id label.
> >
> > (5) *"Figure 2c caption"*
> >
> > Thanks! We have updated the caption with more details.

---

### Review · Reviewer_UV1p · 2023-04-04

**Summary Of Contributions:**

In this work, the authors present the SKILL challenge, whereby distributed agents must learn in a lifelong manner in parallel across a set of sequential tasks. To solve this problem, the authors present a novel method called lightweight lifelong learning (LLL), which obviates issues regarding storage, lack of knowledge transfer, and test-time oracles.

The authors show that existing lifelong learning/continual learning approaches fail to perform well in the SKILL paradigm, and that LLL is able to maintain strong performance on prior tasks, while incurring minimal compute and communication costs. The authors then show evidence that the learned lightweight learners can be combined to facilitate forward transfer.

**Audience:**

Yes

**Claims And Evidence:**

Yes

**Requested Changes:**

* Can the authors provide a better motivation for the SKILL problem setting?
* Can the authors explain why prior baselines perform so poorly in the SKILL-102 benchmark?
* Could the authors provide clarity on the use of teacher-student in their LLL approach v.s. the canonical meaning in deep learning?
* Could the authors update the presentation, as outlined by in the Weaknesses section?

**Strengths And Weaknesses:**

### Strengths
* The paper is fairly well written, and I only needed a few passes to get the key ideas.
* The LLL approach is simple and well motivated, and the authors address a key test-time issue of not having an oracle.
* The authors do a good job of clearly showing the computational and communication costs incurred by their method and baselines, and the baselines shown appear rigorous and fairly evaluated.
* I found the experiments, especially the approach outlined to corrective and forward transfer, quite creative and definitely believe they would be interesting to the wider community.

### Weaknesses
* The SKILL problem setting itself seems a bit contrived; the motivation provided (e.g., object recognition in the sky and the bottom of the ocean) doesn't feel particularly obvious to me. There are motivations to do with resilience, but these don't seem that well explored in the paper.
* I have some concerns over the performance of the baselines. For instance, EWC seems to be remarkably poor, despite being a fairly strong baseline in prior work.
* I'm not entirely sure about the teacher-student nomenclature; teacher-student usually refers to a system whereby we have one neural network distilling knowledge into another, but in this instance, it just seems like this refers to the sharing of statistics for use in test-time task inference, which feels conceptually different. This might be confusing to readers more familiar with the former, especially given Teacher-Student approaches are used already to solve lifelong learning problems [1].
* There are quite a few formatting/presentation issues that should be addressed.
    * In Figure 3, "Knowledge consolidate" is covered by the white space of the "Task mapper" shape (bottom left).
    * The graphs should be vectorized and their presentation improved; Figure 2c is blurry and hard to read. As another example, in Figure 4a., it is very hard to distinguish between the curves as there is so much whitespace between 0 and the worst performing method; the authors should clip the axis range.
    * Figure 4 uses 1th in the title, should use '1st'
    * Similarly, the tables in Figure 9 and 10 should be formatted correctly. At the moment they are just images, so are referred to as figures; however these should be properly formatted LaTeX tables, with a caption above said tables.
    * Figure 12, legend and y-axis labels are squashed.
    * Page 2: "Each LLL agent uses a common frozen backbone built-in at ~manufacturing time~ *initialization*"

[1] Lifelong Teacher-Student Network Learning, Ye and Bors, IEEE Transactions on Pattern Analysis and Machine Intelligence 2022

---

> ### Author Response · Authors · 2023-04-20
> **Feedback for Reviewer UV1p**
>
> Dear reviewer,
>
> Thank you for your detailed and very useful comments, which we have addressed in revising the paper, as described below.
>
> **[Q1 SKILL setting]** we added more concrete motivation scenarios to the revised manuscript (Introduction):
>
> 1) Users each take pictures of landmark places and buildings in their own city, then provide annotations for those. After learning and sharing, all users can identify all landmarks while traveling to any city. This could also apply to recognizing products in stores or markets in various countries, or foods at restaurants worldwide.
> Thus, even though each teacher only learns at one or a few locations (or tasks), eventually all users may be interested in knowledge from all locations, as it will be useful during travel.
>
> 2) Agents in remote outposts worldwide with limited connectivity are taught to recognize symptoms of new emerging diseases, then share their knowledge to allow any agent to identify all diseases quickly.
>
> 3) Explorers are dispatched to various remote locations and each learns about plant or animal species they encounter, then later shares with other agents who may encounter similar or different species.
>
> 4) Each time a criminal of some sorts is apprehended (e.g., shoplifter, insurgent, spy, robber, sex offender, etc), the local authorities take several hundred pictures to learn to identify that person. Then all local authorities share their knowledge so that any criminal can later be identified anywhere.
>
> **[Q2 Baseline performance]**  One reason is that the SKILL-102 benchmark is quite challenging. We also trained EWC on the permuted-MNIST dataset (using 10 permutations) and we find that the performance of EWC is very high (97%). This confirms that EWC works great on easier benchmarks. Cheung et al. 2019 [PSP baseline [1]] also found the same 97% EWC accuracy on permuted MNIST in their Fig. 3. However, when we increase the task complexity, for instance, testing on split CIFAR-100 (20 tasks, 5 classes each), we find the performance drops to <50%. The same finding was reported in the GEM paper [Lopez-Paz et al 2017, [2]], in their Fig. 1. Likewise, in the BPN paper [Wen et al., 2021,[3]], the accuracy of EWC is also only 50% on the “8-dataset” sequence (those datasets are the first 8 in our 102-dataset sequence). That 8-dataset sequence was proposed by Aljundi et al. (2018) [4], who also found in their Fig. 5 that EWC performs at about 50% accuracy after learning those 8 datasets.
>
> We also conduct new experiments on the Visual Domain Decathlon dataset (10 tasks with on average 312 classes/task [Rebuffi et al [5]]). Please see the new suppl. Figure S8. Again, our approach performs much better than the baselines. We posit that this is because these tasks (like ours) are much harder than MNIST, involving higher-resolution natural images, a higher number of classes per task, higher number of training images, higher intra-class heterogeneity, and a much larger domain gap between tasks.
> This is one of the reasons why we want to share our SKILL-102 dataset with the community, as it seems that it is much more challenging than previous datasets and hence could help spur new innovation in LL research.
>
> Reference:
>
> [1] Cheung, Brian, et al. "Superposition of many models into one." Advances in neural information processing systems 32 (2019).
>
> [2] Lopez-Paz, David, and Marc'Aurelio Ranzato. "Gradient episodic memory for continual learning." Advances in neural information processing systems 30 (2017).
>
> [3] Wen, Shixian, et al. "Beneficial perturbation network for designing general adaptive artificial intelligence systems." IEEE Transactions on Neural Networks and Learning Systems 33.8 (2021): 3778-3791.
>
> [4] Aljundi, Rahaf, et al. "Memory aware synapses: Learning what (not) to forget." Proceedings of the European conference on computer vision (ECCV). 2018.
>
> [5] Rebuffi, Sylvestre-Alvise, Hakan Bilen, and Andrea Vedaldi. "Learning multiple visual domains with residual adapters." Advances in neural information processing systems 30 (2017).
>
> **[Q3 Teacher-student]**  Yes, here we want to show that the teacher and student are roles that each agent will perform: Each agent is the "teacher" for its assigned tasks, and the "student" for the other tasks. We agree with your concern, and we have clarified this in the revised version (Sec. 3).
>
> **[Q4 Format and presentation]** Thanks. We have modified them in the revised version:
>
> *Figure 3*: We moved things around a bit but this may be a rendering issue as it looks fine on our machines. We will make sure it renders correctly at press time.
>
> *Fig. 2c and Fig. 4a*: Thanks, we have updated Figure 2c to vectorized now (PDF) and enlarged the font size. We will make sure the quality is as good as possible when preparing the proofs. We also clipped the axis range of Fig. 4a and correct the typo of “1st”
>
> *Fig. 9 and 10, 12*: Thanks! We have changed Fig. 9 and 10 to Tables 1 and 2. We updated new Fig. 10 (used to be Fig. 12).

---

### Review · Reviewer_4LNS · 2023-04-07

**Summary Of Contributions:**

This work proposes a distributed lifelong learning approach, where multiple physically distributed resources (datasets) are available, each with multiple tasks. Therefore, a Lifelong Learning agent (LLL) is assigned to each resource, and each LLL agent is a sequential lifelong learner capable of learning multiple tasks in its region one after the other. Each agent consists of a shared backbone and a specific head, and communication is enabled among agents to share knowledge, allowing each agent to solve all tasks.

**Audience:**

Yes

**Claims And Evidence:**

Yes

**Requested Changes:**

Question:

1. The proposed setting is not commonly seen. Since each location only has data for some tasks, why is it necessary to test on tasks from other locations? Under what circumstances, in reality, is the proposed framework necessary? Could you give some examples?

It seems that the challenges proposed in the paper have not been well addressed:

2. Challenge 1 did not use a global agent, it assumed that all distributed agents can communicate globally, which seems no different.

3. Challenges 2. Which specific module contributed to addressing the lifelong learning challenges? Can you explain if you have designed mechanisms to prevent forgetting previously learned knowledge in lifelong learning? This is not clearly reflected in the paper.

4. Challenge 3. I understand that the knowledge sharing mentioned in the paper refers to the ability of different agents to share their task-specific heads and GMMC through communication. However, in reality, the shared representation is the fixed backbone network. If each agent could extract generalizable knowledge from its own task and update the backbone network, would it achieve better knowledge sharing?

5. Challenge 5. Can synergies occur between different agents? In my understanding, when an agent needs to perform tasks of other agents, it uses the corresponding task-specific head, so this is just an adaptation issue. Can we say that synergies only occur among tasks from one agent? Even so, this synergy mainly comes from the ability of the neural network itself, could you explain which specific module design of this work contributed to harnessing possible synergies among tasks?


**Strengths And Weaknesses:**

Strength：

1.	The paper proposes a new benchmark, SKILL, which fills a gap in the existing benchmarks for lifelong learning and federated learning.

2.	The paper presents comprehensive experimental results to evaluate the proposed method and demonstrate its effectiveness.

Weakness：

1.	This work mainly applies existing methods to a novel setting that combines federated learning with lifelong learning. However, there is a lack of significant new designs proposed to address the challenges of lifelong learning or federated learning. Therefore, the novelty of this work is limited.

2.	The author proposes five challenges that SKILL needs to address, but it seems that the challenges proposed have not been well addressed.

---

> ### Author Response · Authors · 2023-04-20
> **Feedback for Reviewer 4LNS**
>
> Dear reviewer,
>
> Thank you for your insightful comments, which we have addressed in revising the paper, as described below.
>
> **[Q1, New Design]**  *Novel designs for lifelong learning*:
>
> (1) Beneficial Bias (BB): BB enables for the first time using a shared pretrained backbone, as BB can compensate for large domain shifts. Previously, most LL approaches have retrained the whole backbone. This is too expensive to share. Our new BB solves that.
>
> (2) Mahalanobis task mapper: This is a new way to alleviate the need for a task oracle that has plagued much of previous LL research.
>
> *Novel designs for SKILL challenges (​focuses on parallel (speed up) task learning and knowledge sharing among agents)*:
>
> (3) Lightweight Lifelong Learner: To our knowledge, our agent and system design is novel.  It differs from single-agent LL, federated learning, meta-learning, etc in significant ways which we explain in the paper, and also outperforms all baselines in our SKILL challenge.
>
> (4) We propose a synergy learning approach (Sec. 7.2 ) for Lightweight Lifelong Learners, which shows promising forward knowledge transfer, by reusing the accumulated knowledge for faster and more accurate learning of new tasks.
>
> **[Q2, SKILL setting]**  Good point, we added more concrete motivation scenarios to the revised manuscript (Introduction):
>
> 1) Users each take pictures of landmark places and buildings in their own city, then provide annotations for those. After learning and sharing, all users can identify all landmarks while traveling to any city. This could also apply to recognizing products in stores or markets in various countries, or foods at restaurants worldwide.
> Thus, even though each teacher only learns at one or a few locations (or tasks), eventually all users may be interested in knowledge from all locations, as it will be useful during travel.
>
> 2) Agents in remote outposts worldwide with limited connectivity are taught to recognize symptoms of new emerging diseases, then share their knowledge to allow any agent to identify all diseases quickly.
>
> 3) Explorers are dispatched to various remote locations and each learns about plant or animal species they encounter, then later shares with other agents who may encounter similar or different species.
>
> 4) Each time a criminal of some sorts is apprehended (e.g., shoplifter, insurgent, spy, robber, sex offender, etc), the local authorities take several hundred pictures to learn to identify that person. Then all local authorities share their knowledge so that any criminal can later be identified anywhere.
>
> **[Q3, Challenge 1]** It is very different for total failure probability. In our scenario, if we start with 10 agents each learning 10 tasks, then 5 agents are destroyed early during learning of their tasks, then we still end up with 5 remaining agents all mastering 50 tasks. In federated learning, if the central server is destroyed, we end up with 10 remaining agents mastering no task. While the central server might be more resilient, the fact that there is a non-zero probability of ending with nothing is not acceptable in some scenarios.
> Another difference is that we minimize the amount of data shared, which is typically much higher in federated learning (e.g., 91 MBytes of Xception model gradients per task for federated learning vs. 2 MBytes in our approach).
> We added this in related works of Federated Learning.
>
> **[Q4, Challenge 2]**
> Each of our LLL agents is a lifelong learner. This is enabled by their new design: fixed task agnostic backbone + Beneficial Bias + head + task mapper. These components working together enable LL in our agents, that is, each agent can learn a sequence of tasks with minimal forgetting or interference.
> A desirable feature of LL is the ability of forward transfer, which we design and implement for our agents in Sec. 7.
> We now added this in Sec 5, 2nd para.
>
> **[Q5, Challenge 3]** If each agent has their own updated task-specific backbone, then the sharing cost is much higher than ours: sharing only the Beneficial Bias, head, and GMMC/Mahalanobis task mapper. For Xception, sharing the 22.9M weights would require 91 MBytes per task, while we share < 2 MBytes/task, a reduction of 45x.
>
> **[Q6, Challenge 5]** The synergies can occur between different agents as shown in Sec. 7. Here the synergies refer to the ability to reuse learned (received from other agents) knowledge from old tasks to boost the learning speed and accuracy of new tasks. Sec. 7.2 shows the positive knowledge transfer among agents. In short, before training, we could match the new task classes to similar learned (received from other agents) old tasks. If the class similarity is larger than a threshold, we treat it as a matched class, then we use the similar old class weights to initialize the weights of the new class. Otherwise, we use random initialization. Table 1 and Fig. 14 show that our method can reuse knowledge from other agents to achieve better performance and faster convergence.

---

### Decision · Action_Editors · 2023-05-09

**Recommendation:** Accept as is

**Comment:**

I'll be brief since there is not much need to elaborate, given the reviews: the reviewer consensus is that this paper makes a timely and meaningful contribution to the lifelong learning literature, and is of an acceptable standard for publication at TMLR. The reviewers all give fairly clear constructive feedback, which I see has been responded to and incorporated into an updated draft by the authors. I see no reason to further withhold acceptance.

**Audience:**

From my own reading of the paper, and from the reviews, the paper is clearly of interest to the TMLR readership.

**Claims And Evidence:**

Reviewer consensus is that the paper makes a timely and well-supported contribution to the lifelong learning literature, and that barring minor issues which have been addressed, the claims are suitably shored up the experiments.